



# How do extreme ENSO events affect Antarctic surface mass balance?

Jessica M.A. Macha[1], Andrew N. Mackintosh[1], Felicity S. McCormack[1], Benjamin J. Henley[2,3], Helen V. McGregor[3,4], Christiaan T. van Dalum[5], and Ariaan Purich[1]

[1]Securing Antarctica's Environmental Future, School of Earth, Atmosphere and Environment, Monash University, Clayton, Kulin Nations, VIC 3800, Australia
[2]School of Agriculture, Food & Ecosystem Sciences, University of Melbourne, Burnley, VIC 3121, Australia
[3]Securing Antarctica's Environmental Future, School of Earth, Atmosphere and Life Sciences, University of Wollongong, Wollongong, NSW 2522, Australia
[4]Environmental Futures, School of Earth, Atmosphere and Life Sciences, University of Wollongong, Wollongong, NSW 2522, Australia
[5]Utrecht University, Institute for Marine and Atmospheric Research Utrecht, Princetonplein 5, 3584 CC Utrecht, The Netherlands

**Correspondence:** Jessica M.A. Macha (jessica.macha@monash.edu)

**Abstract.** Extreme El Niño-Southern Oscillation (ENSO) events have far-reaching impacts globally, yet their impacts on Antarctica are poorly understood. In particular, how extreme ENSO events influence Antarctica's mass balance remains uncertain, with few studies considering how extreme events could differ from moderate events. Here, we examine the impacts of past extreme El Niño and strong La Niña events over the period 1979–2018 on surface mass balance of Antarctica using

a reanalysis-forced regional climate model. We find that Antarctic surface mass balance does not vary significantly during most of the simulated extreme events. Regional impacts differ between individual events and cannot be generalized across all extreme events. Enderby Land is an exception: significant increases in surface mass balance – approximately 32% of the regional annual average – occur during all extreme El Niño events. Furthermore, during the 2015/16 extreme El Niño event, widespread and significant surface mass balance changes occurred across East and West Antarctic catchments. These changes

are remarkable, extending outside the respective catchments' 5th and 95th probability distributions for September-November period. Our results suggest that future extreme ENSO events may continue to cause significant impacts in Antarctic surface mass balance. However, the magnitude and polarity of the potential impacts cannot be inferred from the limited information available on extremes contained in four decades of historical data. Further investigations using ice core data and large ensemble model simulations are needed to better understand the drivers of the spatial and temporal variability in this system.

## 1 Introduction

The El Niño-Southern Oscillation (ENSO) is a key driver of Antarctic surface climate (Fox-Kemper, et al., 2021). ENSO influences Antarctic climate via an atmospheric teleconnection, whereby Rossby wave trains propagate from the tropical Pacific to the poles (Turner, 2004; Ciasto et al., 2015). The Rossby wave train changes the large-scale atmospheric circulation in the





southern polar latitudes, shifting the strength, position and spatial extent of the Amundsen Sea Low (ASL), with impacts
on Antarctic surface climate (Turner et al., 2013; Raphael et al., 2016; Clem et al., 2017). ENSO is therefore an important
influence on variability in Antarctic surface mass balance (SMB) (e.g. Goodwin et al., 2016), via impacts on polar atmospheric
circulation (e.g. Renwick and Revell, 1999), atmospheric temperature (e.g. Steig et al., 2009) and snowfall patterns (e.g.
Cullather et al., 1996; Guo et al., 2004). These impacts are important because the balance between snowfall-driven SMB gain
and ocean-driven basal melting determine the Antarctic Ice Sheet (AIS) contribution to sea level rise (Meredith et al., 2019;
Huguenin et al., 2024).

Previous studies exploring ENSO-driven Antarctic impacts have considered both moderate and extreme ENSO events to-
gether in analyses (e.g. Welhouse et al., 2016; Clem et al., 2018; Paolo et al., 2018; King et al., 2023; King and Christoffersen,
2024; Macha et al., 2024), but have typically not considered how the impacts of extreme ENSO events differ from moderate
strength events (Cai et al., 2015a; Santoso et al., 2017; Paolo et al., 2018). Extreme ENSO events, defined here as when the
sea surface temperature (SST) anomaly of the central tropical Pacific Niño-3.4 region (5°N–5°S, 170°W–120°W; Trenberth,
1997) exceeds +/- 2°C, are associated with devastating global impacts (Cai et al., 2014, 2015b): the 1997/98 El Niño event
claimed over 20,000 human lives worldwide (Changnon, 1999; Sponberg, 1999); the 1998/99 strong La Niña event flooded
50% of Bangladesh's land area (Ninno and Dorosh, 2001; Mirza et al., 2001) and displaced over 20 million people (Jonkman,
2005).

Studies focusing on the influence of extreme ENSO events on the AIS mass balance are limited (Boening et al., 2012; Paolo
et al., 2018; Bodart and Bingham, 2019). This is partly because only a small number of extreme events have been observed
in the satellite era, limiting the ability to produce a statistically robust "baseline" of present-day Antarctic impacts associated
with these rare events. Paolo et al. (2018) shows that intense El Niño events are associated with net ice shelf mass loss in the
Amundsen/Bellingshausen sector, but does not consider other parts of the ice sheet. Bodart and Bingham (2019) considered
the impacts of the 2015/16 extreme El Niño event and found that it was associated with an unprecedented mass gain in the
Antarctic Peninsula within the GRACE observational period (2002 to 2017). Lowlying coastal East Antarctica experienced a
brief period of reduced net mass loss during this event, associated with enhanced precipitation (Bodart and Bingham, 2019).
However, impacts from the 2015/16 El Niño event have not been compared to those from other extreme El Niño events in
Antarctica and therefore the Antarctic SMB response during this event may not be typical. Furthermore, the ENSO 'flavour'
was different for the 2015/16 El Niño event, which began as a Central Pacific (CP) event, compared to the extreme El Niño
events of 1982/83 and 1997/98, which were Eastern Pacific (EP) events (L'Heureux et al., 2017; Santoso et al., 2017; Lieber
et al., 2024; Macha et al., 2024).La Niña events are associated with strong Antarctic mass balance responses in the Amundsen
sector including reduced Antarctic shelf melting (Huguenin et al., 2024), less snowfall (Sasgen et al., 2010) and reduced ice
shelf height(Paolo et al., 2018). However, these studies do not isolate the influence of strong La Niña events on continent wide
Antarctic SMB.

This study investigates the impact of historical extreme ENSO events on Antarctic SMB. We use the Regional Atmospheric
Climate Model version 2.3p3 (RACMO2.3p3), assessing the temperature, snowfall and SMB anomalies associated with three
extreme El Niño events (1982/83, 1997/98, 2015/16) and the five strongest La Niña events (1988/89, 1998/99, 1999/2000,





2007/08 and 2010/11) from 1979–2018. We consider both regional and continental scales, focusing on each major catchment
of the AIS. Overall, we aim to answer the following questions: do Antarctic impacts from extreme ENSO events follow a
similar pattern? Where do these impacts occur? And more generally, do these extreme ENSO events also result in extreme
Antarctic SMB impacts?

## 2 Methods

### 2.1 Data

We analyse seasonal averages of atmospheric variables across Antarctica (regions and catchments defined in Figure 1), calcu-
lated over: Dec–Feb (DJF), Mar–May (MAM), Jun–Aug (JJA), and Sep–Nov (SON). All anomalies are calculated relative to
the 1979–2018 mean (unless otherwise specified). All fields are linearly detrended prior to analysis.

#### 2.1.1 Regional climate model RACMO2.3p3

We assess Antarctic climate variability using 27km resolution output from the Regional Atmospheric Climate Model version
2.3p3 (RACMO2.3p3) simulated over the period 1979–2018 (van Dalum et al., 2021, 2022). RACMO2.3p3 is forced using
3-hourly output from the European Centre for Medium-Range Weather Forecasts atmospheric reanalysis (ERA5) (Hersbach
et al., 2020; van Dalum et al., 2021). Macha et al. (2024) show that although 1979—1984 is considered a spin up period
in RACMO2.3p3, there is limited impact of excluding this period on the statistical robustness of analyses, and therefore we
include this period in our analyses.

#### 2.1.2 Reanalysis data

To link the teleconnections of extreme ENSO events to their impacts in Antarctica, we use mean Sea Level Pressure (SLP)
from the global reanalysis ERA5 outputs (Hersbach et al., 2020) across the mid-latitudes to polar latitudes (the RACMO2.3p3
simulation only extends across the polar latitudes). ERA5 outputs are provided at 0.25° resolution, and we utilise the outputs
for the entire region south of 45°S latitude, capturing atmospheric variables across southern mid-latitudes to the southern polar
latitudes (Hersbach et al., 2020). We calculate seasonal anomalies by removing the mean over over the period 1979–2018, to
enable direct comparison with the RACMO2.3p3 variables.

#### 2.1.3 Niño-3.4, $N_{CP}$ and $N_{EP}$ El Niño-Southern Oscillation indices

ENSO events are defined according to the location and magnitude of peak SST anomalies in the tropical Pacific (Trenberth,
1997; Ashok et al., 2007; Kug et al., 2009). In this study, we use three ENSO indices — namely, Niño-3.4, $N_{CP}$ and $N_{EP}$ —
to account for diversity in the location of peak SST anomalies, and hence "flavour" of ENSO events.

Niño-3.4 SST anomalies are calculated from the average equatorial SST across the tropical Pacific from 5°N–5°S, and
170°W–120°W according to Trenberth (1997); available from https://climatedataguide.ucar.edu/ climate-data/nino-sst-indices-



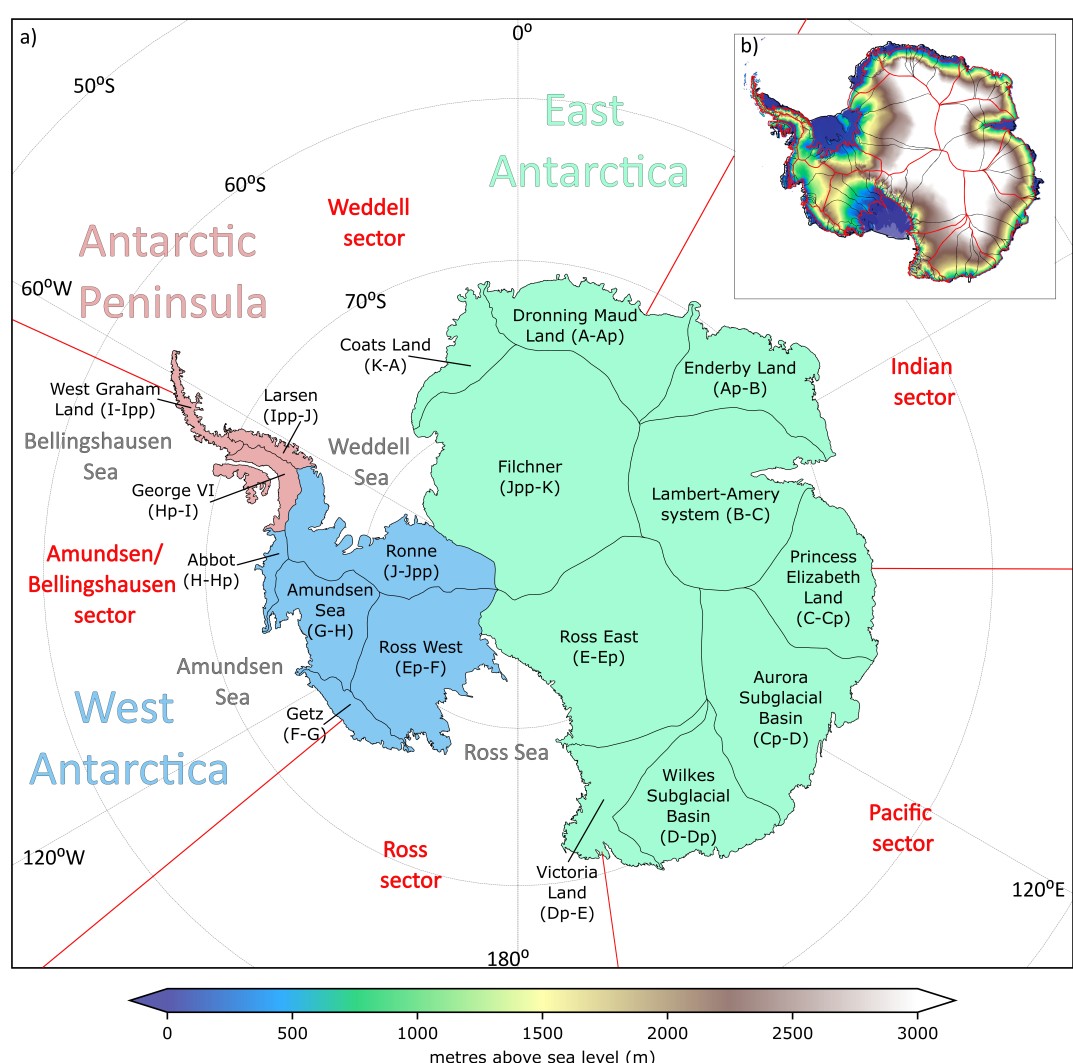

**Figure 1.** The AIS (a) showing subregions and catchment names used in this study, as defined by Rignot et al. (2011)

and Rignot et al. (2019). The Rignot et al. (2011) catchment naming system, based on lettered regional catchments, is included in brackets.

Antarctic Peninsula (pink), East Antarctica (green) and West Antarctica (blue) areas are highlighted. Weddell, Amundsen/Bellingshausen,

Ross, Indian and Pacific sectors are shown in red. (b) Inset Antarctic map of Antarctica surface elevation (contours; metres above sea level)

from Liu et al. (2001), with individual catchments (red lines) and catchment sub-regions (black lines) from Rignot et al. (2011) and Rignot

et al. (2019).





nino-12-3-34-4-oni-and-tni, (last accessed: 3 May 2024). Here, SST anomalies are sourced from the National Oceanic and

Atmospheric Administration (NOAA) (Rayner, 2003), based on the Hadley Centre Sea Ice and Sea Surface Temperature

dataset (HadISST). We use the Trenberth (1997) definition of extreme El Niño and La Niña events, namely when the SST
anomaly of the central tropical Pacific Niño-3.4 region exceeds +/- 2°C for six months or more.

Extreme ENSO events are classified as CP or EP-type events in the same way as Macha et al. (2024) according to the Ren
and Jin (2011) $N_{CP}$ and $N_{EP}$ indices, respectively (see supplementary text S1). Ren and Jin (2011) $N_{CP}$ indices are also used
to identify non-extreme CP-type events in the time period which are used for comparative purposes in our analysis (see section

90 2.1.4).

### 2.1.4 Defining extreme ENSO events

Over 1979–2018 three extreme El Niño events occurred: 1982/83, 1997/98 and 2015/16. The 1982/83 and 1997/98 events
are classified as EP El Niño events (Figure 2c) and are the strongest EP El Niño events on record (L'Heureux et al., 2017;
Santoso et al., 2017). By contrast, the 2015/16 extreme El Niño event was classified as a CP-type El Niño event, but as the

event matured, warming in the eastern Pacific became stronger than in the central Pacific making the event difficult to classify
(Santoso et al., 2017; Xue and Kumar, 2017). The 2015/16 extreme El Niño event did not yield as high SST anomalies in the
EP region as the 1982/82 and 1997/98 events; however, it was associated with the largest SST anomaly in the Niño-3.4 region
(L'Heureux et al., 2017; Santoso et al., 2017; Xue and Kumar, 2017). In this study, we consider each of these three events
separately. Non-extreme CP El Niño events (see section 2.1.3) are also included in our analysis, to provide a comparison to the

impacts associated with extreme El Niño events, building off Macha et al. (2024), which found that CP-type El Niño events
result in widespread SMB increases in West Antarctica.

No extreme La Niña events with an SST anomaly <-2°C occur between 1979–2018 (Figure 2), as La Niña events are
associated with smaller amplitude SST anomalies than El Niño events (Capotondi et al., 2015; Lieber et al., 2024). Therefore
in this study we define strong La Niña events as occurring when the 3-month average Niño-3.4 SST anomalies are < -1.5°C

for six months or more from 1979–2018. Between 1979–2018 five strong La Niña events occur: 1988/89, 1998/99, 1999/2000,
2007/08 and 2010/11. Strong La Niña events typically occur the year following a strong El Niño event (Takahashi et al., 2011),
and the 1998/99 and 1999/2000 events are sequential, following the extreme 1997/98 El Niño event (Figure 2).

El Niño SST anomalies generally initiate between March and June, and develop in JJA and SON, reaching their peak in
DJF, whilst peak La Niña SST anomalies generally occur in SON (Webster, 1982; Ambrizzi et al., 1995; Trenberth, 1997).

In this study, we focus on ENSO impacts in SON (of the year when the ENSO event develops), when the ENSO-Antarctic
teleconnection is strongest (Webster, 1982; Ambrizzi et al., 1995; Lee et al., 2009), due to a strong subtropical jet enabling
Rossby waves to propagate from the tropics to the poles (Renwick and Revell, 1999; Turner, 2004; Yiu and Maycock, 2020).
By contrast, the ENSO-Antarctic teleconnection is weakest during DJF, so whilst extreme El Niño events typically peak during
DJF, we do not include periods beyond SON in our main analysis (Webster, 1982; Renwick and Revell, 1999). The analysis of

other seasons and annual results is included in the supplementary material (Supplementary Figures S1-S3, S9, S10).





**Figure 2.** Time series of anomalies of (a) total SMB for the AIS (black; from RACMO2.3p3, (van Dalum et al., 2021); (b) Niño-3.4 SST anomalies (from NOAA (Rayner, 2003)), (c) CP El Niño index ($N_{CP}$; yellow) and (d) EP El Niño index ($N_{EP}$; green) (Ren and Jin, 2011). Extreme El Niño events (pink) and strong La Niña events (blue) are highlighted across all time series.



## 2.2 Analyses

### 2.2.1 Defining & aggregating Antarctic regions

We analyse the RACMO2.3p3 SMB fields from the 18 individual catchments used in (IMBIE) Antarctic mass balance assessments (Figure 1; Rignot et al., 2011), to consider regional SMB changes. Total SMB is aggregated by catchment to allow
comparison between catchments.

### 2.2.2 Cumulative SMB anomalies

Seasonal cumulative regional SMB anomalies are calculated across all grid cells within a given catchment, and over all months within the season, allowing the total monthly-scale SMB extremes to be included (Webster, 1982; Ambrizzi et al., 1995; Lee et al., 2009). Cumulative annual SMB anomalies are the summed 12-month regional SMB anomalies, allowing the contribution
of ENSO events relative to the yearly SMB budget to be explored.

### 2.2.3 Statistical analyses

We undertake analysis on the SON composites of all extreme El Niño and strong La Niña events SLP, 2-metre air temperature, precipitation and SMB anomalies. We also compare SLP, 2-metre air temperature, precipitation and SMB anomalies for each individual extreme El Niño event and strong La Niña event. Statistical significance is tested using a Students' $t$ test or a
two-sided Kolmogorov-Smirnov (K-S) test at the 5% confidence level, as indicated in Figure captions.

We first assess the probability distribution of cumulative SON SMB anomalies for each catchment and year, delineating extreme ENSO events from other events. We also define extreme SMB changes as occurring when anomalies exceed the 5th or 95th percentile and identify outliers in each region's cumulative SON SMB anomaly dataset (see supplementary Text S2 for outliers methodology).

We assess whether SMB anomalies during extreme ENSO events deviate from baseline conditions by comparing probability distributions, as follows. First, we determine a "baseline" probability distribution by calculating the 90th and 95th percentiles for each catchments' SON SMB anomalies over the full period from 1979–2018. Second, we calculate the probability distribution of SMB anomalies for each catchment during extreme El Niño events, CP El Niño events, and strong La Niña events. Third, we compare the probability distribution of SMB anomalies during these extreme events with the "baseline" probability
distributions previously calculated, testing differences in distributions for statistical significance. Finally, we compare results for extreme and moderate ENSO events, to place our findings in the context of ENSO variability, as well as comparing SON results with annual results to place our findings in the context of the annual surface mass budget.

Statistical significance of differences in distributions are tested using Monte-Carlo Sampling (1000 simulations) with a two-sided Kolmogorov-Smirnov (K-S) test at the 5% confidence level (see Supplementary Tables S1-S4; Hammersley and
Handscomb, 1964; Smirnov, 1948). We test:





– if SMB anomalies during each (i) extreme El Niño event, (ii) CP El Niño event, and (iii) La Niña event, are statistically different from baseline conditions (Supplementary Table S1, Supplementary Table S3); and

– if SMB anomalies distributions between each extreme ENSO event are statistically distinguishable from one another (Supplementary Table S2 and S4).

All results presented in the main figures are statistically significant, unless otherwise indicated by stippling for non-significant results. Auto-correlation is not accounted for in the analysis as the seasonally-averaged lag-one auto-correlation values for 2-metre temperature, precipitation and SMB variables are below 0.15 (for each variable's corresponding units) and are therefore not significant (Mudelsee, 2010).

## 3   Results

### 3.1   Spatial maps of Antarctic surface climate responses to extreme/strong ENSO events

### 3.1.1   Extreme El Niño events

We first consider the composite extreme El Niño behaviour by averaging the individual extreme El Niño events of 1982/83, 1997/98 and 2015/16. Extreme El Niño events are on average associated with a positive SLP anomaly in the Amundsen/-Bellingshausen sector, a negative SLP anomaly in the Ross Sea and a negative SLP anomaly in the Weddell Sea (Figure 3a).
This results in a 'dipole' in temperature anomalies across West Antarctica, with positive temperature anomalies across the Amundsen/Bellingshausen and Ross sectors and negative temperature anomalies across the Weddell Sea (Figure 3b). A weak positive pressure anomaly also occurs over Enderby Land (Figure 3a).

  We next consider whether the composite extreme El Niño behaviour is consistent across all the individual events. During all three extreme El Niño events we also see a positive SLP anomaly in the Amundsen/Bellingshausen sector and negative SLP
anomalies in both the Ross and Weddell sectors (Figure 3e, i, m). However, in East Antarctica we see differing SLP patterns (Figure 3e, i, m). During the 1982/83 event, the positive SLP anomaly in the Amundsen/Bellingshausen sector is shifted northwards, with the negative SLP anomaly in the Weddell sector incurring further over the continent than the composite (Figure 3e). In the 1997/98 El Niño event, a positive SLP anomaly occurs in the Amundsen/Bellingshausen sector which is of much greater magnitude than during other El Niño events, being as large as 8hPa near the George VI catchment (Figure
3i). This SLP extends far into the interior of the continent and across both East and West Antarctica, in contrast to the other extreme El Niño events (Figure 3i). During both the 1982/83 and 1997/98 events, as in the composite, positive SLP anomalies occur over Enderby Land (Figure 3e, i). However, during 1997/98 this SLP over Enderby Land is of much greater magnitude than during other El Niño events and is part of the SLP system that extends across the entire continent (Figure 3i). During the 2015/16 event, West Antarctic SLP anomalies are similar to the composite, but a negative SLP anomaly extends from Wilkes
Subglacial Basin across the Pacific sector, with no corresponding positive SLP anomaly over Enderby Land (Figure 3m).

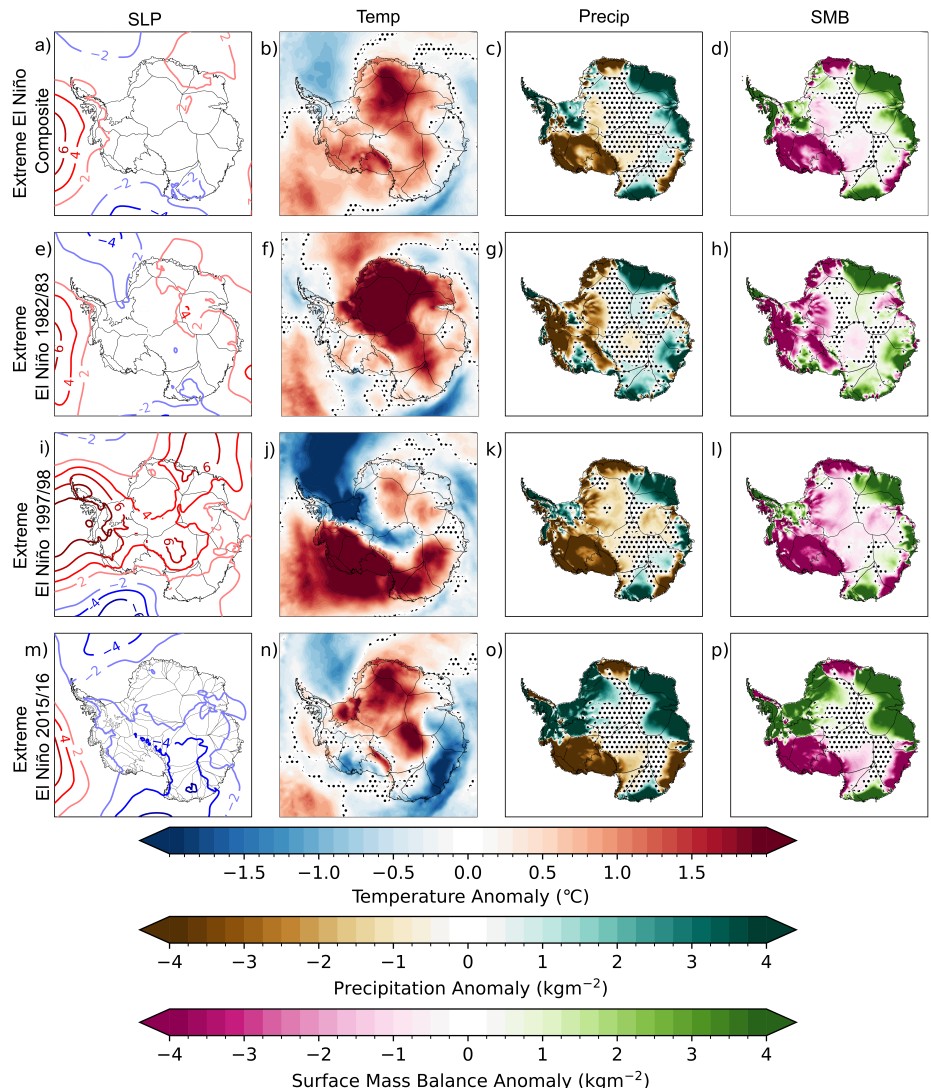

**Figure 3. SON Antarctic surface climate changes during extreme El Niño events.** SON composite (a-d) and SON average (e-p) sea level pressure (SLP; coloured contours at 2hPa increments; from ERA5; first column); Antarctic 2-metre atmospheric temperature (Temp, from RACMO2.3p3; second column); precipitation (precip, from RACMO2.3p3; third column) and SMB (from RACMO2.3p3; fourth column) during extreme El Niño events (a-d) in 1982/83 (e-h), 1997/98 (i-l) and 2015/16 (m-p).Fields are linearly detrended prior to analysis (see Methods). Non-statistically significant results at the 5% confidence level using a two-tailed Students' *t* test are shown with stippling.



These SLP circulation features result in similar positive temperature anomalies during the 1982/83 and 2015/16 El Niño events across the Weddell and Indian sectors, with positive anomalies extending inland across the Filchner catchment and into the Ross East catchment (Figure 3f, 3n). Negative temperature anomalies are also similar, occurring in the Pacific Ocean during both events, but during the 1982/83 event this anomaly is constrained off the Antarctic coast, whilst during the 2015/16 event the negative temperature anomaly extends over the Pacific sector of East Antarctica across Wilkes and Aurora Subglacial Basins (Figure 3f, n). During the 1997/98 event, these circulation changes result in positive temperature anomalies in the Amundsen/Bellingshausen sectors, and a negative anomaly extending into the Weddell sector and inland East Antarctica (Figure 3j).

Precipitation and SMB anomalies are concentrated predominantly along the coast, and across West Antarctica during El Niño events (Figure 3). Positive precipitation and SMB anomalies in the Indian sector during each El Niño event are consistent with the composite pattern; however, other regions do not have consistent responses during different El Niño events (Figure 3g-h, k-l, o-p). Precipitation and SMB anomalies in the Weddell sector and parts of the Ross sector contrast with those of the composite pattern during the 1982/83 event, with negative anomalies across much of the Weddell sector and positive anomalies across parts of the Ross sector (Figure 3g-h). During the 1997/98 event, precipitation and SMB anomalies are generally consistent with the extreme El Niño composite pattern across the continent, except that weak negative anomalies extend further inland in the Filchner catchment (Figure 3c-d, k-l). During the 2015/16 event, large positive precipitation and SMB anomalies in the Weddell sector and across the Peninsula are in stark contrast to the composite and other El Niño events (Figure 3o-p). Whilst precipitation and SMB anomalies in the Weddell sector differ to the composite during both the 1982/83 and 2015/16 events, these responses differ to one another with negative precipitation and SMB anomalies during the 1982/83 event and widespread large positive anomalies during the 2015/16 event (Figure 3g-h, o-p). Precipitation and SMB anomalies in the Antarctic Peninsula, Weddell sector, Aurora Subglacial Basin, Princess Elizabeth Land and Lambert-Amery System catchments vary considerably between individual extreme El Niño events, with extreme El Niño events being associated with both negative and positive anomalies in these regions (Figure 3g-h, k-l, o-p). This highlights the lack of a consistent mass balance response to extreme El Niño events across Antarctica (Figure 3).

### 3.1.2 Strong La Niña events

Strong La Niña events induce a range of impacts, with no clear pattern in common between strong La Niña event surface climate anomalies (Figure 4). The composite of the five strong La Niña events shows a negative SLP anomaly in the Amundsen/Bellingshausen sector, indicating a strengthened ASL, resulting in negative temperature anomalies in the Ross sector and positive temperature anomalies in the Weddell sector (Figure 4a-b). This strengthened ASL is also seen in four of the five strong La Niña events, with no strengthened ASL being evident during the 2007/08 event – unusual for a La Niña event (Figure 4e, i, m, q, r, u). However, the magnitude and extent of this negative SLP anomaly as well as the resultant temperature anomalies varies between events (Figure 4f,j, n, v). During the 1988/89 event, a positive SLP system extends across the continent, and positive anomalies extend across the Peninsula, Weddell and Ross sectors (Figure 4e-f). In contrast, during the 1998/99 event negative SLP anomalies occur in the Amundsen/Bellingshausen sector and over Enderby Land catchment, whilst negative tem-



perature anomalies extend across West and East Antarctica except over the Aurora Subglacial Basin (Figure 4i-j). Furthermore, during the 1999/2000 and 2010/11 events negative SLP systems extend across the continent, and positive temperature anomalies extend varying amounts across the Antarctic Peninsula (Figure 4m-n, u-v). In East Antarctica, there is no clear pattern between temperature anomalies during strong La Niña events (Figure 4f,j, n, r, v).

Regional precipitation and SMB anomalies also vary considerably between the events, with numerous regions experiencing
negative and positive anomalies during different strong La Niña events (Figure 4g-h, k-l, o-p, s-t, w-x). Whilst precipitation and SMB anomalies are concentrated predominantly along the coast, and across West Antarctica during strong La Niña events (Figure 4), during the 2007/08 event positive temperature anomalies extend into the Filchner catchment and Pacific sector of East Antarctica, causing positive precipitation and SMB anomalies inland in the Filchner catchment compared with the composite (Figure 4c-d, s-t). This highlights the lack of consistent mass balance response to La Niña events across Antarctica
(Figure 4).

### 3.2  Magnitude of catchment-scale surface mass balance changes during extreme/strong ENSO events

We next consider whether SMB responses at the catchment scale are extreme (i.e., outside the 5th or 95th percentiles) in the context of all SMB changes during the 1979–2018 period. We find that there are no consistent responses between extreme El Niño events (comparing red triangles in Figure 5) or strong La Niña events (comparing blue triangles in Figure 5), except in
Enderby Land, East Antarctica (Figure 5). Here, the three extreme El Niño events all have cumulative SON SMB anomalies at approximately 8,000kg/m$^2$ (Figure 5). In fact, the largest positive regional cumulative SMB anomalies in Enderby Land between 1979 and 2018 (other than in 2009, which our analysis identified as an outlier) occur during these extreme El Niño events (Figure 5). During La Niña events, we do not see evidence of a consistent SMB response in Enderby Land, with only the 1988/89 La Niña event SON SMB anomaly exceeding the 5th percentile at <-5000kg/m$^2$ (Figure 5).

In all other catchments, the magnitudes of the SMB anomalies during extreme/strong ENSO events are not consistently extreme, but in numerous catchments a single extreme/strong ENSO event is associated with an extreme SMB response (Figure 5). In the Wilkes Subglacial Basin and George VI catchment, the 2015/16 extreme El Niño event is associated with the largest positive cumulative SMB anomalies, at approximately 12,500kg/m$^2$ and 11,000kg/m$^2$, respectively (Figure 5). However, the other two extreme El Niño events are associated with SMB anomalies within interquartile ranges (between 2,500kg/m$^2$ and
-3,000kg/m$^2$, and 3,000kg/m$^2$ and -4,500kg/m$^2$, respectively) (Figure 5). In Princess Elizabeth Land the 2007/08 La Niña event is associated with the largest positive cumulative SMB anomaly in the region for the 1979–2018 period at 9,000kg/m$^2$, but other strong La Niña events SMB anomalies are within the regions' interquartile range (Figure 5). In Getz and Amundsen Sea catchments, SMB anomalies during 1998/99 and 1999/2000 strong La Niña events are the largest positive cumulative SMB anomaly in the time period around the 95th percentile at approximately 6,000kg/m$^2$ and 11,000kg/m$^2$, respectively (Figure 5).

As explained in section 2.1.3, we compare extreme El Niño events to non-extreme CP El Niño events as Macha et al. (2024) show that CP El Niño events are associated with larger SMB responses than EP events. In the Aurora Subglacial Basin, significant SMB changes do not occur in any extreme El Niño event (anomalies are within the interquartile range, between -4,500kg/m$^2$ and 4,500kg/m$^2$), but three CP El Niño events are associated with large positive anomalies around the 95th



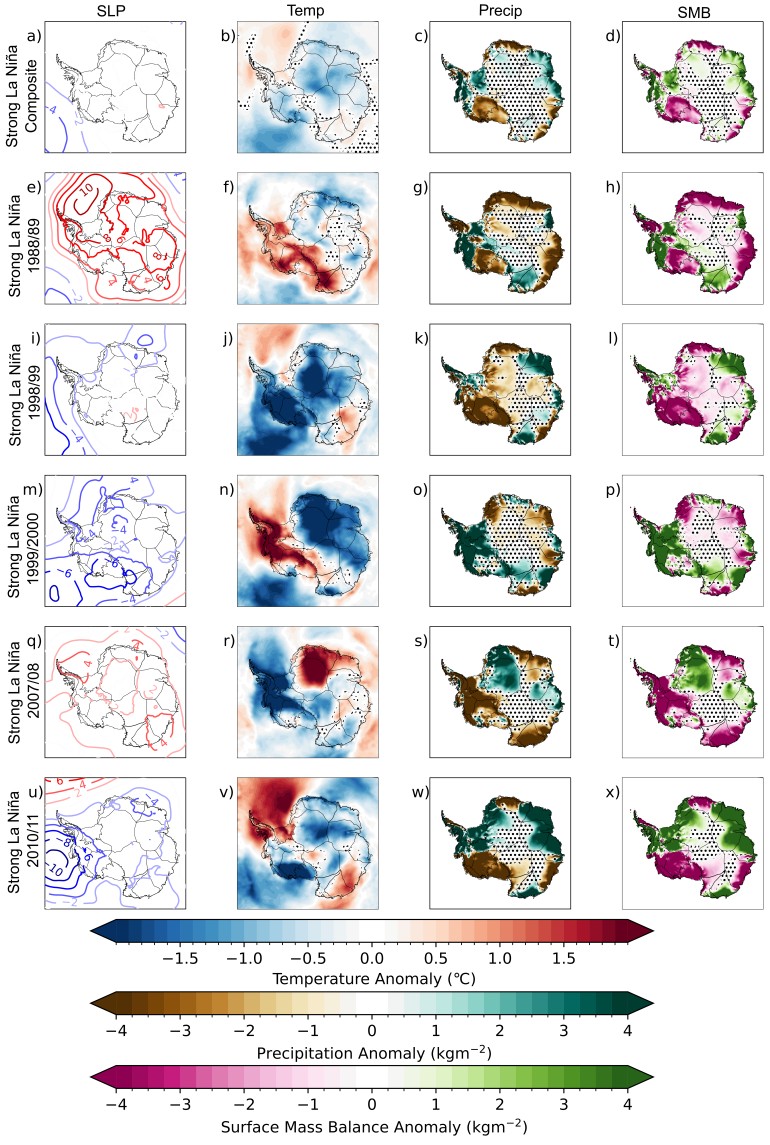

**Figure 4. SON Antarctic surface climate changes during strong La Niña events.** SON composite (a-d) and SON average (e-x) sea level pressure (SLP; coloured contours at 2hPa increments; from ERA5; first column); Antarctic 2-metre atmospheric temperature (Temp, from RACMO2.3p3; second column); precipitation (precip, from RACMO2.3p3; third column) and SMB (from RACMO2.3p3; fourth column) during strong La Niña events (a-d) in 1988/89 (e-h), 1998/99 (i-l), 1999/2000 (m-p), 2007/08 (q-t) and 2010/11 (u-x). Fields are linearly detrended prior to analysis (see Methods). Non-statistically significant results at the 5% confidence level using a two-tailed Students' *t* test are shown with stippling.





**Figure 5. Relationship between extreme ENSO events and regional Antarctic surface mass balance anomalies in SON.** Density curves of regional cumulative SON SMB anomalies for each Antarctic Ice Sheet regional catchment (a-j). Violin plots are scaled (x-axis of violin plots) by the regional catchment size, meaning each violin width is proportional to the size of each catchment. Box plots show the interquartile range, with medians (black line) and whiskers (5th and 95th percentiles). Catchments are ordered according to Rignot et al. (2011), starting at Dronning Maud Land and working clockwise around the Antarctic coastline. East Antarctic (light green), West Antarctic (light blue) and Antarctic Peninsula (pink) catchments, outliers (crosses; see supplement), extreme El Niño events (red), strong La Niña events (blue) and Central Pacific El Niño events (yellow) are highlighted. Scatter plots of regional cumulative SON SMB anomalies plotted against the Niño-3.4 index for all catchment regions are included in the supplement (Supplementary Figure S4).





percentile, between 9,000kg/m$^2$ and 12,000kg/m$^2$ (Figure 5). In the Getz catchment, two strong La Niña events (1998/99 and

1999/2000) are associated with positive SMB anomalies outside the 95th percentile; however, the other strong La Niña events are associated with SMB anomalies within the regions' interquartile range (Figure 5). In the Ross West region, two CP El Niño events (1991/92 and 2002/03) are associated with approximately 10,000 kg/m$^2$ and 16,000 kg/m$^2$ respectively, with all other SMB anomalies being <6,000kg/m$^2$ (Figure 5). For other catchments, responses vary by event, with no consistent or significant SMB changes (Figure 5).

Distinct responses to extreme ENSO events are also seen during other seasons (Supplementary Figure S1-S3). For example, large positive cumulative SMB anomalies are identified during summer for different strong La Niña events in Dronning Maud Land (2010/11), Enderby Land (1988/89), the Lambert-Amery System (1999/2000) and Princess Elizabeth Land (2007/08; Supplementary Figure S1). This highlights that the magnitude of the catchment-scale mass balance changes we identify during extreme/strong ENSO events are not necessarily extreme (Figure 5; Supplementary Figure S1-S3).

**3.3    Statistical significance of SMB anomalies during extreme/strong ENSO events**

Finally, we consider whether strong/extreme ENSO events are significantly different from moderate ENSO events in terms of SON SMB anomalies. Figures 6-8 show that there is no clear pattern indicating that strong/extreme ENSO events result in extreme SON SMB anomalies. That is, significant SON SMB response to individual extreme El Niño events are evidenced in numerous individual catchments (Figure 6). However, these changes are not consistent between events, with the 'centres of

action' for increased SMB occurring in different catchments during each El Niño event in the record (Figure 6). Enderby Land is an exception, with statistically significant SON SMB increases occurring during all three extreme El Niño events (Figure 6a). Enderby Land's average anomaly during these events is equivalent to approximately 32% of the regions' annual average SMB (Figure 6a).

During the 2015/16 El Niño event, significant SMB changes are widespread across Antarctica: in Enderby Land, Wilkes

Subglacial Basin, Victoria Land, Ross East, George VI and Ronne catchments (Figure 6a, c, d, e, g, h). The 2015/16 SMB probability distribution is shifted positively in all these catchments, except for in the Ross East catchment, where the SMB probability distribution is shifted negatively (Figure 6a, c, d, e, g, h). During the other two extreme El Niño events studied, significant SMB changes only occur in Enderby Land (1982/83 and 1997/98, both probability distributions are shifted positively; Figure 6a) and the Ross West catchment (1997/98; probability distribution is shifted negatively; Figure 6f). In all other

catchments, SMB changes during extreme El Niño events are not significant (Figure 6b, Supplementary Figure S7).

Significant SON SMB responses during CP El Niño are localised (in <5 grid cells; Figure 7). In the Aurora Subglacial Basin, Victoria Land, Ross East and Ross West catchments, a shift in the SMB probability distribution during a CP El Niño event (2002/03 or 2009/10 dependent upon catchment) is larger than the SMB response during the 1982/83, 1997/98 and 2015/16 events (Figure 7b, d, e, f, S6). Most other catchments do not show significant results during CP El Niño events (Figure 7a, c,

g, h). Regional results with each extreme El Niño event and CP El Niño event distinguished are included in Supplementary Figure S6. Extreme El Niño and CP El Niño results are also significant when compared with the SMB anomaly distribution of moderate El Niño events in most catchments (Supplementary Figure S8). This highlights that the SMB changes we identify





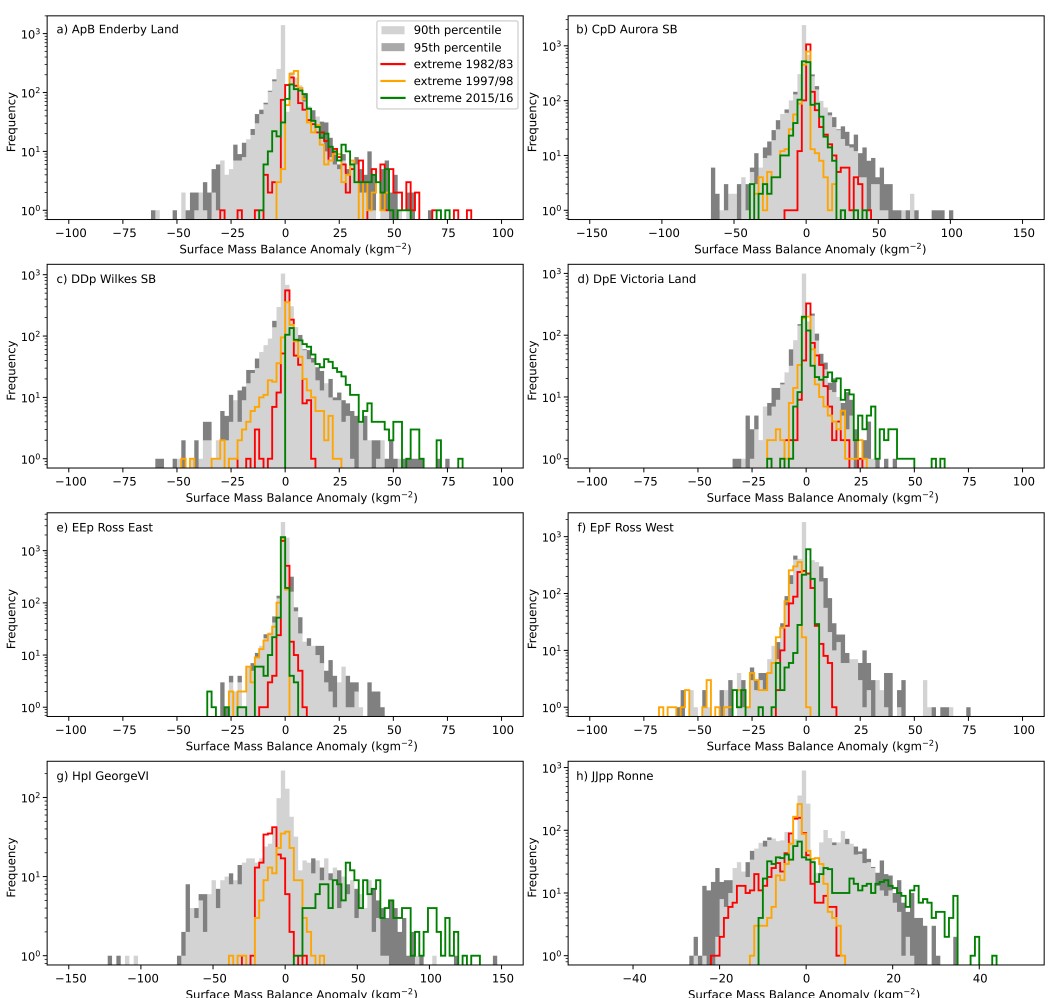

**Figure 6. Comparing probability distributions of regional Antarctic surface mass balance anomalies during extreme El Niño events.** Probability distributions of SON SMB anomalies in Antarctic catchments with significant changes in SON during extreme El Niño events (1982/83 (red), 1997/98 (yellow) and 2015/16 (green), and regional 90th (light grey shading) and 95th percentile (dark grey shading) regional SMB anomalies for SON for 1979–2018 period. Only statistically significant regional results are shown at the 5% confidence level using a two-sided Kolmogorov-Smirnov (K-S) test and Monte Carlo Sampling (see Methods). Full results and two-sided Kolmogorov-Smirnov (K-S) tests are included in the supplement (Supplementary Figures S6-S7).




during extreme or CP El Niño events are not necessarily statistically significantly different to those of moderate El Niño events (Figure 6-7; Supplementary Figure S1-S3).

Few Antarctic catchments exhibit significant SMB anomalies during strong La Niña events (Figure 8, Supplementary Figure S10). Similar SMB changes occur during moderate and strong La Niña events (Supplementary Figure S12). Significant SMB changes during strong La Niña events only extend beyond the moderate La Niña SMB anomaly distribution in the Wilkes Subglacial Basin, Victoria Land and Ross East catchments (Supplementary Figure S11). This highlights that the mass balance changes during strong La Niña events across almost all of Antarctica are not extreme when compared to those of moderate La Niña events (Figure 6-7; Supplementary Figure S1-S3).

Annual SMB anomalies during extreme El Niño events are less significant than their SON equivalent (Supplementary Figure S9). Annual SMB anomalies during strong La Niña events are similar to those in SON (Supplementary Figure S12). This highlights that the contribution of extreme/strong ENSO events relative to the yearly SMB budget is not necessarily extreme or significant in many Antarctic catchments.

## 4 Discussion

Here, we have considered how extreme ENSO events in the historical record impact Antarctic SMB to determine if extreme ENSO events are associated with extreme SMB changes in Antarctica. We find that the 2015/16 extreme El Niño event was associated with more widespread and significant SMB changes than other extreme ENSO events. This is consistent with previously noted impacts on Antarctica (Bodart and Bingham, 2019), and that 2015/16 is the most extreme El Niño event over the past 40 years (Bell et al., 2015).

However, beyond the 2015/16 event, our results show that extreme ENSO events do not necessarily produce extreme impacts in Antarctic climate and SMB. That is, we identify differences in the spatial patterns and magnitude of event-to-event responses. Whilst regional changes in SLP, temperature, precipitation and SMB have occurred during extreme ENSO events, anomalies during these events are not statistically different from 'baseline' conditions or moderate ENSO events. Enderby Land is an exception, and exhibits a consistent response to the three extreme El Niño events in Antarctica in SON; all other catchments do not. These findings are supported by the considerable variability in ENSO, with differing teleconnections between events (Ciasto et al., 2015; Yuan et al., 2018; Yiu and Maycock, 2020).

### 4.1 Extreme ENSO event impacts and moderate ENSO event impacts

In many Antarctic catchments, the SMB response differs greatly between the ENSO events that we examined. We find that the magnitude and spatial patterns of these differences are so great that these impacts are not consistent between events and the composite is not representative. When comparing the impacts of extreme and moderate ENSO events, we find that numerous catchments exhibit differing SMB responses. Moderate and extreme El Niño event SMB changes are significantly different from one another in Enderby Land, Wilkes Subglacial Basin, Victoria Land, Ross East, Ross West, Getz, Amundsen Sea, Abbott, George VI and Ronne catchments. However these results are not consistent (other than in Enderby Land) across



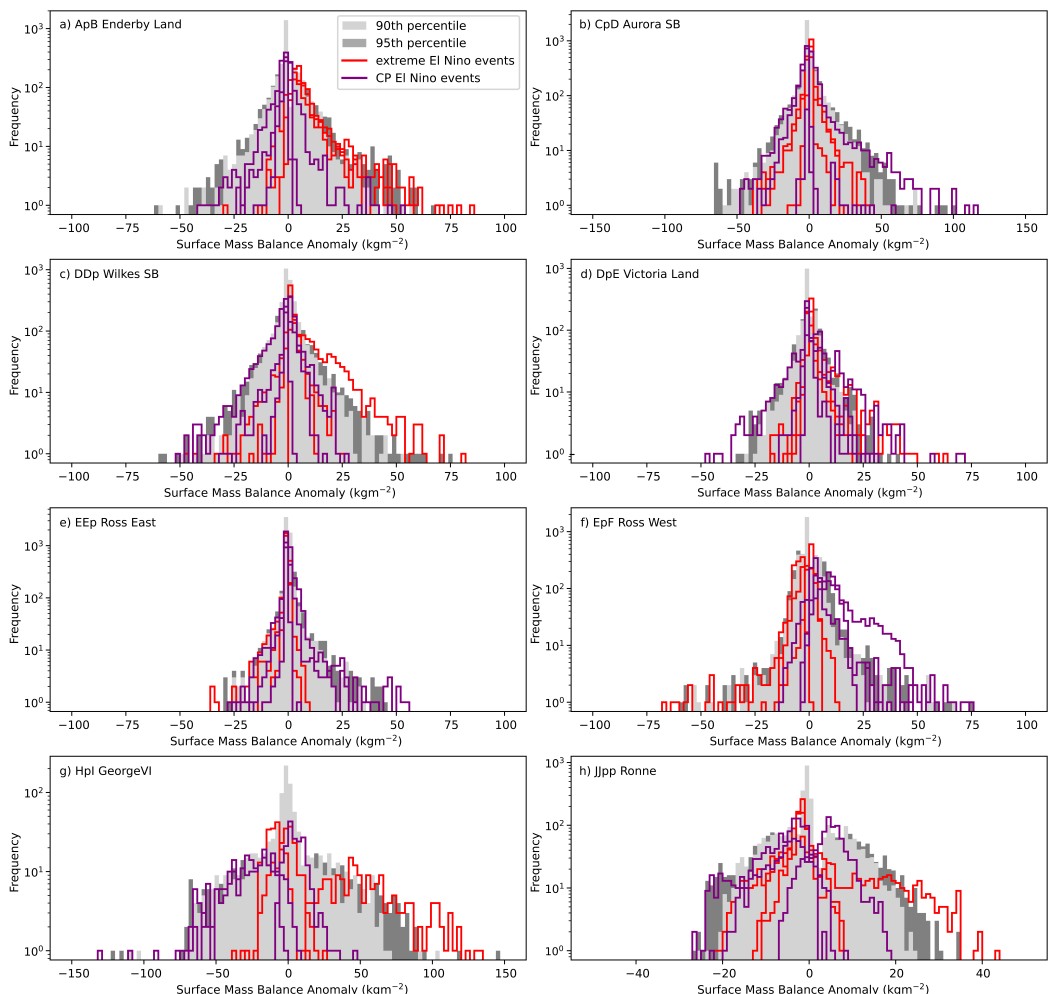

**Figure 7. Comparing probability distributions of regional Antarctic surface mass balance anomalies during extreme El Niño events and CP El Niño events.** Probability distributions of SMB anomalies in Antarctic catchments with significant changes in SON for extreme El Niño events (red lines) and CP events (purple lines), and regional 90th (light grey shading) and 95th percentile (dark grey shading) SMB anomalies for SON for 1979–2018 period. Only statistically significant regional results are shown at the 5% confidence level using a two-sided Kolmogorov-Smirnov (K-S) test and Monte Carlo Sampling (see Methods). Two-sided Kolmogorov-Smirnov (K-S) test results are included in the supplement. Only regions with significant SMB changes are shown in the main text, SMB histograms for all other catchments are included in the supplement (Supplementary Figures S3).



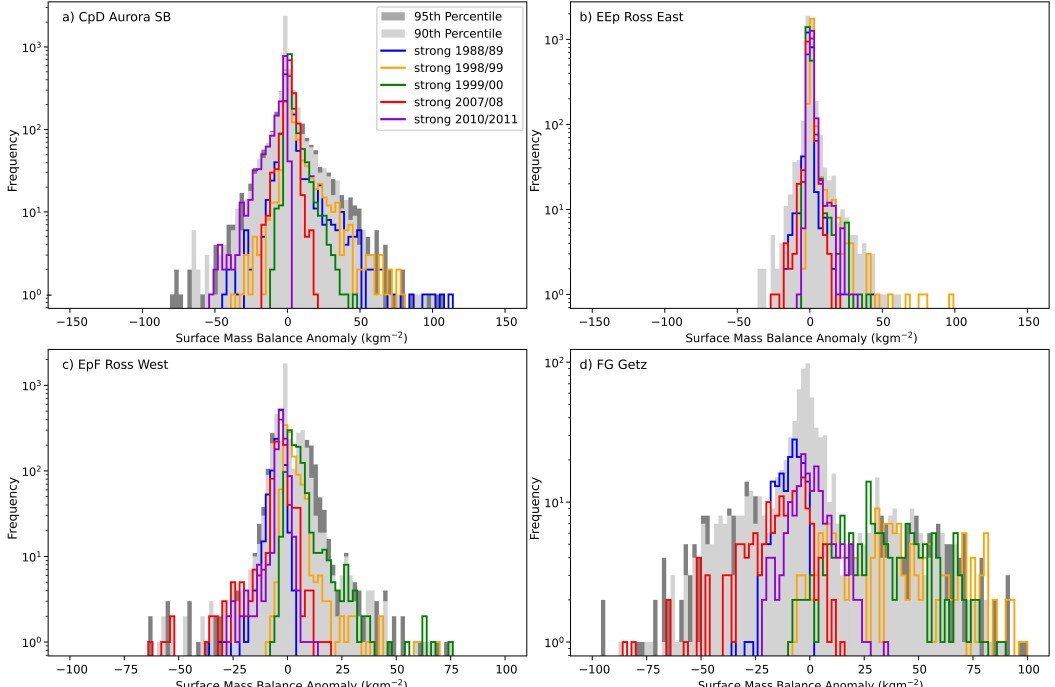

**Figure 8. Comparing probability distributions of regional Antarctic surface mass balance anomalies during strong La Niña events.**
Probability distributions of SMB anomalies in Antarctic catchments with significant changes in SON for strong La Niña events (blue lines), in 1988/89 (sea blue), 1998/99 (teal), 1999/2000 (cyan), 2007/08 (navy) and 2010/11 (sky blue), and regional 90th (light grey shading) and 95th percentile (dark grey shading) SMB anomalies for SON for 1979-2018 period. Only statistically significant regional results are shown at the 5% confidence level using a two-sided Kolmogorov-Smirnov (K-S) test and Monte-Carlo Sampling (see Methods). Two-sided Kolmogorov-Smirnov (K-S) test results are included in the supplement. Only regions with significant SMB changes are shown in the main text, SMB histograms for all other catchments are included in the supplement (Supplementary Figure S10).

extreme El Niño events. Santoso et al. (2017) and Srinivas et al. (2024) suggest that such findings relate to the non-linear nature of extreme ENSO events, driven by the convective response to extreme tropical Pacific SST anomalies and resultant Rossby wave propagation.

Conversely, numerous other catchments exhibit similar SMB responses, that is, the impacts of extreme ENSO events do not differ from those of moderate events (including in Dronning Maud Land, Lambert-Amery system, Princess Elizabeth Land,

Aurora Subglacial Basin, West Graham Land, Larsen, Filchner and Coats catchments). These findings support studies that have considered moderate and extreme ENSO event impacts together in analyses (e.g. Clem et al., 2018; Paolo et al., 2018; King et al., 2023), which suggests that the Antarctic response is not sensitive to the magnitude of ENSO events.

Smaller Antarctic SMB changes occur during strong La Niña events than extreme El Niño events. This may be partially explained by the smaller amplitude and weaker Antarctic teleconnection of La Niña events when compared to El Niño events,



which mean these events are often also associated with smaller impacts (Capotondi et al., 2015; Srinivas et al., 2024). Understanding the differences in the teleconnection dynamics between El Niño and La Niña events is an important area of research, but investigating the remote mechanisms of influence is beyond the scope of this study.

## 4.2 Extreme El Niño impacts in Enderby Land

Our findings support previous work showing that snowfall increases in Enderby Land during El Niño events (Reijmer and
Broeke, 2003; Schlosser et al., 2008; Simon et al., 2024). We identify a SLP system and anomalous atmospheric circulation over the Indian sector of East Antarctica during each extreme El Niño event (Figure 3e-f, i-j, m-n). This circulation change causes warm onshore winds to extend into the Indian sector of Antarctica during these events, bringing moisture-laden air over the region, resulting in increased precipitation and SMB. Each El Niño event is associated with slightly different spatial patterns in SLP and temperature as the circulation system is shifted east or west around the coastline. This spatial variability results in
moisture-laden air, precipitation and positive SMB anomalies reaching different catchments during different events. Enderby Land is associated with SMB increases during each of these events as it is in the 'centre of action' of this SLP anomaly.

During the 1997/98 event, we find a large positive SLP anomaly extends across the Indian sector, over Enderby Land and the eastern edge of Dronning Maud Land catchment (Figure 3m). This results in onshore moisture-laden winds over the region and subsequent precipitation (Figure 3m-o), causing SMB increases across the entire Enderby Land catchment (extending inland),
and the eastern edge of Dronning Maud Land catchment (Figure 3p). Our results agree with Boening et al. (2012), who showed a SLP anomaly caused blocking over the Dronning Maud Land and Enderby Land catchments during the 1997/98 event, which enabled storm tracks to extend to Dome Fuji (inland Enderby Land) and brought sustained snow accumulation to the region.

During the 1998/99 and 2010/11 strong La Niña events we also identify negative SLP anomalies and associated increases in precipitation and SMB over Enderby Land (Figure 4i, u). Boening et al. (2012) show that SLP anomalies occur over the
Enderby Land catchment during the 2010/11 strong La Niña event, resulting in sustained precipitation along the Indian sector of the East Antarctic coastline. Our findings support Boening et al. (2012), and highlight that similar SLP systems and blocking processes could occur in the Ross and Amundsen/Bellingshausen sectors, depending upon the ASL location, as well as in the Weddell and Pacific sectors, where SLP anomalies are identified during La Niña events in our analysis.

These findings also highlight that El Niño and La Niña event impacts are not always reciprocal (Capotondi et al., 2015;
Lieber et al., 2024). Therefore, if this teleconnection were to persist under future climate change, we expect that more frequent and intense ENSO extremes (no matter the polarity) will result in snow accumulation increases in Enderby Land.

Snow accumulation in inland Enderby Land is partially driven by short-term, high-magnitude precipitation events, which extend inland a few times each year (Reijmer and Broeke, 2003; Schlosser et al., 2008, 2010; Simon et al., 2024). These events develop due to the large-scale circulation, which facilitates the development of atmospheric rivers that funnel warm,
moist air across the southern mid-latitudes to Antarctica, resulting in high, sustained precipitation (Turner et al., 2019; Wille et al., 2021, 2024). The sheer volume of snowfall during extreme Antarctic precipitation events can, in a matter of days, offset regional melting (Davison et al., 2023; Wille et al., 2024). Future research should consider the interactions between teleconnection dynamics and atmospheric river development, determining if ENSO events facilitate such extensive snowfall.





### 4.3 The 2015/16 El Niño event

The 2015/16 El Niño event was associated with significant SMB changes across Antarctica, consistent with previous research focusing only on this event (Bodart and Bingham, 2019). Our findings further show that the 2015/16 event impacts in Antarctica stand out relative to previous events, with more widespread and significant SMB changes than other extreme ENSO events. One reason for this may be because the 2015/16 extreme El Niño event displayed CP-type characteristics (Santoso et al., 2017; L'Heureux et al., 2017), which typically enhance snow accumulation in Antarctica compared with EP El Niño events (Macha

et al., 2024).

In the Wilkes Subglacial Basin, the 2015/16 extreme El Niño was associated with SON SMB anomalies approximately 25% larger than during any other year from 1979–2018. The 2015/16 event was the most extreme El Niño event during the past 40 years, with a larger SST anomaly magnitude than the 1982/83 and 1997/98 events (Santoso et al., 2017; L'Heureux et al., 2017). This magnitude difference was partially attributed to the 2015/16 event being initiated from a warmer tropical Pacific

background state than the 1982/83 and 1997/98 events (Santoso et al., 2017), resulting in the higher magnitude and more widespread Antarctic impacts.

We find no apparent anthropogenic trend in the magnitude of the extremes in our analyses (Supplementary Figure S5). Some studies have suggested that the larger magnitude SST anomaly of the 2015/16 El Niño event could indicate a change in ENSO behaviour due to climate change (e.g. Xue and Kumar, 2017; Cai et al., 2015b)). Furthermore, a previous study

(Medley and Thomas, 2019) could not rule out an anthropogenic driver of recent regional increases in snow accumulation. Attributing whether there is an anthropogenic signal in these extreme ENSO impacts is beyond the scope of this study and requires centennial scale datasets to fully characterise ENSO variability (Stevenson et al., 2010).

### 4.4 Other drivers & future outlook for SMB

This study focuses on extreme ENSO events; however, we have not considered other climate variability including the South-

ern Annular Mode (SAM), which is known to influence the Antarctic climate and interact with ENSO (Fogt and Marshall, 2020; Medley and Thomas, 2019). For example, during the 1982/83 and 1997/98 extreme El Niño events, the SAM was in a negative phase, with weaker circumpolar westerly winds around Antarctica (Marshall, 2003). But during the 2015/16 El Niño event, the SAM was in a positive phase, leading to stronger and poleward shifted westerly winds (Marshall, 2003; Bodart and Bingham, 2019). These differing circumpolar wind anomaly patterns could explain why positive precipitation and SMB

anomalies extended further inland during the 2015/16 event than the 1982/83 and 1997/98 events, as stronger moist, onshore winds, contracted around the continent, combined with orographic uplift could have resulted in enhanced precipitation and SMB gain further inland. Schlosser et al. (2010) and Fogt et al. (2011) have shown that the advection of warm air from lower latitudes towards East Antarctica is facilitated during the negative phase of SAM and has been linked to ENSO-driven circulation changes, resulting in enhanced precipitation in regions such as Enderby Land (Noone et al., 1999; Reijmer and Broeke,

2003; Schlosser et al., 2008, 2010). Whilst we do not consider the SAM in our analyses, our results are consistent with these





findings. A recent study (King et al., 2023) shows that the SAM explains up to 24% of Antarctic ice-mass trends since 2002, highlighting that the SAM almost certainly exerts an influence on our results and extreme El Niño and strong La Niña events.

Our analysis of the Antarctic mass balance impacts from extreme ENSO events is limited by the length of the datasets available. In this study we utilise the full 40-year (1979–2018) temporal extent of RACMO2.3p3 (van Dalum et al., 2021).

However, only a handful of extreme ENSO events occur in this period (three El Niños and five La Niñas). This period also includes both natural climate variability and anthropogenic forced change, which we are unable to separate due to the limited time series length of this analysis (Meredith et al., 2019; Fox-Kemper, et al., 2021). Adequately characterising the statistics of ENSO requires a record on the order of one-to-two centuries long (Wittenberg, 2009; Stevenson et al., 2010) and this is not currently available. Future work utilising palaeoclimate reconstructions and climate model large ensemble simulations would

increase the sample size and may reduce sampling bias (Wittenberg, 2009; Stevenson et al., 2010). Despite these limitations, our examination of SMB responses during the satellite era provides useful and preliminary insights into the impact of extreme ENSO events on Antarctica.

Projections of future ENSO characteristics remains uncertain, including how the teleconnection between the tropics and the poles could change under future climate change (Fox-Kemper, et al., 2021; Freund et al., 2024). Some CMIP6 model outputs

suggest that extreme ENSO events may become more frequent in the future (Cai et al., 2023; Lieber et al., 2024). It is important to understand how Antarctica will respond in a future climate that is warmer and experiences greater extremes (Meredith et al., 2019). Our results show that extreme ENSO events do not result in large SMB changes across Antarctica. Whilst we do identify consistent and significant SMB changes in Enderby Land, this is the result of atmospheric circulation changes specific to this region. Therefore, whether increases in extreme ENSO events in the future could lead to significant impacts on Antarctica's

surface climate is dependent upon how the teleconnection to Antarctica changes (McGregor et al., 2022; Cai et al., 2023). If future El Niño events were to resemble the 2015/16 El Niño event, the resulting increased accumulation could offset some of the ocean-driven melting in Antarctica, reducing mass loss (Huguenin et al., 2024).

## 5   Conclusions

We use reanalysis-forced regional climate model output alongside reanalysis data to quantify the impact of the largest ENSO

events of the late 20th and early 21st centuries on Antarctica's surface mass balance. We show that the Enderby Land catchment is associated with a consistent and significant increase in SMB during all extreme El Niño events over the satellite record. The average anomaly during these largest events (1982/83, 1997/98 and 2015/16) is equivalent to approximately 32% of that regions' annual average surface mass balance. In all other Antarctic catchments, the surface mass balance changes differ greatly between the ENSO events that we examined. When comparing the impacts of extreme and moderate ENSO events, we find

that numerous catchments exhibit similar SMB responses; that is, based on our analysis over the satellite era, the Antarctic SMB response to ENSO does not seem to be sensitive to the magnitude of ENSO events. However, the 2015/16 extreme El Niño event stands out and is associated with widespread and significant surface mass balance changes. Our results suggest that extreme ENSO events have not had an extreme impact on Antarctic surface mass balance for most catchments.



*Code and data availability.* All datasets are freely available. Regional Atmospheric Climate Model version 2.3p3 output two-metre atmo-
spheric temperature, precipitation and surface mass balance are available from van Dalum et al. (2021) and online at https://doi.org/10.5281/zenodo.7639053.
The Python code used to undertake this analysis can be accessed online at https://figshare.com/s/1b86fdb69b217721e84b and will be pub-
lished upon acceptance. ERA5 reanalysis data including mean sea level pressure are available from Hersbach et al. (2020) and online at
https://doi.org/10.1002/qj.3803. Catchment basins are available from Rignot et al. (2011). SST anomalies to calculate El Niño indices Niño-
3.4, Niño-3 and Niño-4 are available from Rayner (2003) and online at https://doi.org/10.1029/2002JD002670. Supplementary Material is
available online at https://figshare.com/s/1d160c0fa30f34e0189d and will be published upon acceptance.

*Author contributions.* JM conceived the concept of the study, performed the analysis and wrote the manuscript. All other authors provided
guidance at each stage, and were included in discussions, with each author contributing to editorial review of the manuscript.

*Competing interests.* Felicity S McCormack is an editor of The Cryosphere.

*Acknowledgements.* JMAM, ANM, FSM, AP, HVM and BJH were supported by the Australian Research Council Special Research Initiative
for Securing Antarctica's Environmental Future (SR200100005). JMAM was supported by the Monash Graduate Scholarship (MGS) and
Monash International Tuition Scholarship (MITS). FSM was supported by an Australian Research Council Discovery Early Career Research
Award (DE210101433). CTVD was supported under Horizon 2020 (PROTECT grant no.869304).



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
