# Peer review of "How do extreme ENSO events affect Antarctic surface mass balance?"

_EGUsphere, 2024_

## Author Comment (AC1)

January 2025

**Authors response to reviews on '*How do extreme ENSO events affect Antarctic surface mass balance?*' submitted to *The Cryosphere.**

Masashi Niwano
Editor, The Cryosphere

Thank you for considering our revised manuscript.

We are pleased that both reviewers found our paper to be interesting and clear. We thank the reviewers for their helpful feedback, comments and suggestions which will improve the manuscript.

Here, we provide responses to each of the reviewers' comments. Reviewer comments are in *italics.* Our responses are included in regular text. When noting our changes, we refer to both line numbers in the original manuscript and include wording changes in blue text and quotation marks (" ").

Jessica Macha and co-authors

**Response to Reviewer 1:**

**General comments:**

***Comment on composite analysis:*** *This study uses a regional atmospheric model to investigate the impact of extreme ENSO events on the surface mass balance (SMB) of the Antarctic ice sheet. The detailed SMB change in many subregions and catchments is described quantitatively about the simulated results. While such information is of great value, the robustness of the conclusion is rather limited due to the smallness of the sample size (as the authors are well aware). More specifically, the conclusion drawn from the composite analysis of only three samples (for El Nino) is not highly convincing.*

**Response:**

We thank the reviewer for their constructive feedback and are glad that the manuscript contributes value to understanding ENSO impacts in Antarctic SMB.

We are aware of the limitations of our analysis due to a small sample size of extreme ENSO events and the resultant limitations of the robustness of conclusions (see Lines 388 - 397). We have addressed this – to the extent possible – by expanding our analysis to include strong and moderate El Niño events (see response to reviewer 2 comment 3), and show that increasing the sample size of ENSO events does not change our results.

The reviewer raises the point that the "*conclusion drawn from the composite analysis of only three samples (for El Niño) is not highly convincing*". One of the main aims of including

composite analysis in our manuscript is because it is a common approach to understanding ENSO impacts (e.g., Welhouse et al. 2016). However, a key finding of our analysis is that the composite analysis cannot be used to generalise the behaviour of extreme El Niño or La Niña events. We show this in the results for extreme El Niño (Figure 3a-d) and La Niña (Figure 4a-d) events, highlighting that the composite misses regional differences between events, as well as the spread of different magnitudes and sign changes during different extreme events (Figure 3e-p and Figure 4e-x; lines 163-175). We propose to add further text to the manuscript to explain these limitations:

L128: "We test whether there is a consistent response during all extreme El Niño events or strong La Niña events by comparing individual extreme event results with the composite of the extreme El Niño events or strong La Niña events. If each event is associated with similar behaviour, there should be a consistent response across both the composite and each individual event, and vice versa if there are differences between the composite and each individual event."

L175: "The SLP composite for extreme El Niño events is therefore limited, missing regional signals that are only identified when each individual event is compared (Figure 3)."

L183: "The temperature anomalies identified in the composite are therefore not a clear representation of the temperature anomalies identified during each individual extreme event (Figure 3)."

L199: "Changes in SLP, temperature, precipitation and SMB across Antarctica are not consistent between the 1982/83, 1997/98 and 2015/16 events, with regional differences in the magnitude and sign of anomalies (Figure 3e-p). The impacts of extreme El Niño events in Antarctica therefore cannot be generalised; the extreme El Niño composite results miss key regional differences in impacts during events (Figure 3a-d). Extreme El Niño event impacts therefore need to be compared on a case-by-case basis."

L220: "There are few similarities between surface climate anomalies, including SLP, surface temperature, precipitation and SMB, during strong La Niña events (Figure 4). The strong La Niña composite results miss key regional differences in the sign and magnitude of climate impacts during individual events (Figure 4). This highlights the importance of considering each event and its impacts individually. "

L298: "Whilst regional changes in SLP, temperature, precipitation and SMB occur during extreme ENSO events, these changes often differ between events, and anomalies during these events are not statistically different from baseline conditions or moderate ENSO events."

***Comment on ENSO mechanism/teleconnection and other internal variability:*** *Such a weakness of statistical power can be augmented by revealing the mechanism: why each ENSO event has a differing impact. Is the SMB difference between cases due to the difference in ENSO itself (and its teleconnection) or the effect of other internal variabilities near Antarctica? Neither the effect of other internal variabilities nor the difference of each ENSO event is analyzed in depth. I think the analysis needs to be strengthened to provide more robust insight into interpreting the results.*

**Response:** The reviewer's comment concerns why each extreme event impact differs due to:
1) ENSO mechanism / teleconnection; or the
2) Role of internal variability and other climate variability.

**ENSO mechanism / teleconnection**
It is well-established that El Niño and La Niña events vary in character between events (Capotondi et al. 2015; Timmermann et al. 2018) and previous work has focused on differences in extreme events (Cai et al. 2014; L'Heureux et al. 2017; Santoso et al. 2017; Xue and Kumar 2017). Previous studies have also demonstrated that the teleconnections between diverse ENSO events differ markedly, leading to different impacts in Antarctica (e.g., Clem et al., 2018; Bodart and Bingham, 2019; Zhang et al., 2021; Clem et al., 2022; Macha et al., 2024). Our results are therefore unsurprising: we expect different extreme ENSO events will be associated with different impacts in Antarctica.

However, whether the SMB difference between cases is due to the difference in ENSO and its teleconnection is outside the scope of our analysis. That is, the aim of this study is to quantify the impacts of extreme ENSO events on SMB. Whether these SMB differences are due to the difference in ENSO or other climate drivers is the focus of a separate study that we are currently undertaking. That separate study is a substantial piece of ongoing work with distinct aims from this study.

Ascertaining whether the SMB difference between cases is due to the differences in the ENSO teleconnection requires analysis of output from an ensemble of global climate models and/or pacemaker experiments in order to isolate the Rossby wave train teleconnection, which is not otherwise evident in a single simulation. This is because the impacts of extreme ENSO events on Rossby wave train propagation from the tropics to the poles cannot be isolated from local and regional atmospheric circulation and climatic changes in the southern midlatitudes. We therefore have limited scope to comment on causation in this manuscript.

As the purpose of the present manuscript is to compare the impacts between extreme events, including to understand how and where they differ compared with background and average conditions, we present the climatic SON anomalies in circulation (SLP) and associated fields (temperature, precipitation and SMB) during each individual ENSO event in Figure 3 and 4, and do not attribute these changes to the influence of an extreme ENSO event alone.

To make this clearer, we propose to modify text as follows in Line 129-133: "Composite anomalies during extreme ENSO events and during each individual El Niño and La Niña event are tested for statistical significance against the climatological baseline conditions, which incorporate all climate variability; therefore, the changes presented are not the result of ENSO alone."

**Other internal variability**

The reviewer is correct - we do not analyse the effect of internal variabilities in this study. A detailed examination of synoptic internal variability that may result from the seasonal-scale ENSO teleconnection and Rossby wave train, including local scale changes and internal variabilities, is beyond the scope of this study. This is because the Rossby wave train propagates south in a matter of days, resulting in short term atmospheric circulation changes that cannot be separated from other atmospheric circulation on a monthly or seasonal-scale. However, we have included SON Rossby wave analysis during extreme ENSO events (see response to reviewer 1 comment 3b), which helps visualise this teleconnection, but it does not allow the internal variability to be isolated.

Other modes of climate variability, such as the Southern Annular Mode and zonal wave 3, are also not included in this study. The impacts of these other modes of variability are outside the scope of our analysis, which focuses entirely on the impacts of extreme ENSO events. As mentioned above, a separate piece of work will focus on how different modes of climate variability, including ENSO and the Southern Annular Mode, contribute to Antarctic surface climate changes, and how these contributions can be disentangled and compared. This separate study encompasses a substantial piece of ongoing work.

We recognise the role of other climate variability and local scale variability in our analysis, as discussed in section 4.2 (lines 324-353) and section 4.4 (lines 274-288). We propose to amend these sections as follows:

L352: "Local and regional variability in circulation can therefore play an important role in SMB."

L374: "This study focuses on extreme ENSO events; however, we have not considered other climate variability that influence Antarctic climate on interannual, decadal and interdecadal timescales (Fox-Kemper, et al., 2021). One such mode of climate variability is the Southern Annular Mode , which is known to influence the Antarctic climate and interact with ENSO (Fogt et al., 2020; Medley & Thomas, 2019). For example, during the 1982/83 and 1997/98 extreme El Niño events, the SAM was in a negative phase, with weaker circumpolar westerly winds around Antarctica (Marshall et al., 2003), but during the 2015/16 El Niño event, the SAM was in a very strong positive phase, leading to stronger and poleward-shifted westerly winds (Marshall et al., 2003; Vera & Osman et al., 2018; Bodart & Bingham, 2019). These differing circumpolar wind anomaly patterns, alongside ENSO-driven anomalous atmospheric circulation, could explain some of the differences in precipitation and SMB anomalies we identify during the 1982/83, 1997/98 and 2015/16 events (Reijmer et al., 2003; Schlosser et al., 2010). A recent study (King et al., 2023) shows that the SAM explains up to 24% of Antarctic ice-mass trends since 2002, highlighting that the SAM almost certainly exerts an influence on our results and extreme El Niño and strong La Niña events. Interestingly, the combination of an extreme El Niño event and positive phase SAM in 2015/16 was unusual (Fogt et al., 2020), and this 2015/16 El Niño event was also substantially warmer than previous extreme El Niño events, resulting in unusual regional impacts in southern South America and the Antarctic Peninsula (Vera and Osman et al., 2018). Here, we find the positive precipitation and SMB anomalies do extend further inland during the 2015/16 event than during the other two extreme El Niño events. This is somewhat unexpected, given that other studies suggest increased poleward moisture

transport during negative phase SAM, as weakened westerly winds enable the advection of synoptic systems over the continent, bringing precipitation (Schlosser et al., 2010; Fogt et al., 2011), and in the austral spring of 2015 there was a positive SAM. Whether these anomalies may have been influenced by the warmer temperatures in 2015/16 – as a warmer atmosphere can hold more moisture and thus could be expected to transport it further inland – should be investigated."

*Specific comments:*

**Comment 1 (unclear phrasing in abstract):** *Abstract, L.8-9: It is unclear how much of the observed anomaly is due to the ENSO during 2015/16. Other modes of variability can affect the SMB, and thus the result may not necessarily represent the ENSO response alone. Therefore, I am not sure if the statement of L.11-12 has its evidence within the manuscript.*

**Response:** We thank the reviewer for pointing this out. We propose to rewrite these sentences to clarify the result:

L8-12: "Furthermore, for the 2015/16 El Niño, surface mass balance changes across Antarctic catchments extend beyond the 5th and 95th probability distributions for September-November averages for the full 1979 to 2018 period – much further inland than during other extreme El Niño events – suggesting these changes are not consistent with background conditions."

We also propose to improve the clarity of text in L374: "This study focuses on extreme ENSO events; however, we have not considered other climate variability that influence Antarctic climate on interannual, decadal and interdecadal timescales (Fox-Kemper, et al., 2021)."

L376: "These differing circumpolar wind anomaly patterns, alongside ENSO-driven anomalous atmospheric circulation, could explain some of the differences in precipitation and SMB anomalies we identify during the 1982/83, 1997/98 and 2015/16 events (Reijmer et al., 2003; Schlosser et al., 2010; Fogt et al. 2020). A recent study (King et al. 2023) shows that the SAM explains up to 24% of Antarctic ice-mass trends since 2002, highlighting that the SAM almost certainly exerts an influence on our results and extreme El Niño and strong La Niña events."

**Comment 2 (unclear phrasing in conclusion):** *Conclusion, L.413-414: I did not understand that "SMB changes differ greatly between the ENSO events" on the one hand and "numerous catchments exhibit similar SMB responses when comparing the impacts of extreme and moderate ENSO events" on the other. How are they consistent? Does this indicate that the observed feature does not represent ENSO response? It is misleading to conclude that the SMB response (signal) to extreme and moderate ENSO events is similar if noise and noise are compared (or if the signals are embedded by noise due to the small sample size).*

**Response:** We agree that these sentences are confusing. We propose to rewrite these sentences as follows to be clearer:

L410-416: "Based on our analysis, the Antarctic SMB response to ENSO events does not seem to be sensitive to the magnitude of ENSO events, except in Enderby Land and during the 2015/16 event. We show a consistent and significant increase in SMB over Enderby Land during all extreme El Niño events over the satellite record. The annual average anomaly during the largest events (1982/83, 1997/98 and 2015/16) is equivalent to approximately 32% of the annual average surface mass balance in Enderby Land. In all other Antarctic catchments during all three extreme El Niño events over the satellite record the surface mass balance changes differ between individual events. The 2015/16 extreme El Niño event stands out and is associated with widespread and significant surface mass balance changes. Hence, our results suggest that extreme ENSO events have not had an extreme impact on Antarctic surface mass balance for most catchments over the satellite era."

**Comment 3a (mechanism):** *Without the presented mechanism, it is unclear whether the differences between ENSO events are due to the diversity of ENSO itself or whether they are affected by other internal modes of variability.*

**Response**: We have not discussed causation in this study as isolating the mechanism responsible for these changes would require extensive new modelling analyses (e.g. climate model ensembles and/or pacemaker experiments) to determine and attribute causation associated with solely an ENSO event (as per our response to the main comment above). Rossby wave analysis (see next comment) can help visualise this teleconnection, but it does not allow the mechanism to be isolated.

In Figures 3-8, we are careful to not attribute these changes to ENSO alone, but recognise the role of other climate variability and local scale variability. To make this clearer, we propose to amend the manuscript as follows:

L129: "Composite anomalies during extreme ENSO events and during each individual El Niño and La Niña event are tested for statistical significance against the climatological baseline conditions, which incorporate all climate variability; therefore, the changes presented are not the result of ENSO alone."

**Comment 3b (Rossby wave analysis)**: *It would be helpful to present wave trains from the tropics to the Antarctic so that the link to each ENSO event becomes more visible.*

**Response:** This is an excellent suggestion and we have undertaken this Rossby wave analysis.

Unfortunately, the Rossby wave analysis does not allow the full range of mechanisms underpinning our results to be revealed because ENSO-driven Rossby wave trains cannot be easily isolated from local and regional atmospheric circulation and climate changes in the southern midlatitudes (Renwick and Revell, 1999; Clem et al., 2018; You and Maycock et al., 2020; McGregor et al., 2022). However, the probability distribution analysis (Figures 6-8) does highlight how extreme some of these SMB changes are when compared with the background and average conditions.

We plan to present the Rossby wave propagation during each extreme ENSO event in supplementary figures S1 and S2. This allows the link between the tropics and Antarctica during each ENSO event to be examined and compared, and helpfully visualises this teleconnection relationship:

[Figure]

**Supplementary Figure S1.** Tropical-Polar teleconnections during extreme El Niño events. Austral Spring (SON) 500-hPa geopotential height anomalies (contours) during each extreme El Niño event (a) 1982/83, (b) 1997/98, (c) 2015/16, and associated equatorial Pacific warming (colour shading, bar) with arrows showing Rossby wave propagation schematically.

[Figure]

**Supplementary Figure S2.** Tropical-Polar teleconnections during strong La Niña events. Austral Spring (SON) 500-hPa geopotential height anomalies (contours) during each extreme El Niño event (a) 1988/89, (b) 1998/99, (c) 1999/00, (d) 2007/08, (e) 2010/11, and associated equatorial Pacific cooling (colour shading, bar) with arrows showing Rossby wave propagation schematically.

We propose to include text as follows:

L127: "We undertake analysis of the SON Rossby wave propagation from the tropics to the poles for each extreme El Niño and strong La Niña event examined to illustrate the tropical-polar teleconnection (Supplementary Figures S1-S2)."

L299: "Rossby wave analysis of each extreme ENSO event also shows differences between events (Supplementary Figure S1-S2; Supplementary Text S5). However, we do not include in-depth analysis of the difference between each extreme ENSO event development. This is because other work has previously compared extreme ENSO events in greater depth, including ENSO formation and development, as well as event diversity and intensity (Cai et al. 2014; L'Heureux et al. 2017; Santoso et al. 2017; Xue and Kumar 2017)."

In the supplement we propose to include a section providing mode detail about the Rossby wave analysis, L137-154: "Rossby wave analysis of the austral spring (SON) 500-hPa geopotential height anomalies across the southern hemisphere is undertaken during each extreme ENSO event analyzed (Supplementary Figures S1-S2). This analysis allows the teleconnection between the tropics and the poles to be visualised during each extreme ENSO event, as the propagation pathway is highlighted. This analysis also allows the differences in this propagation between individual extreme ENSO events to be compared. During each El Niño event we note differences in where the Rossby wave train extends over the Antarctic continent, with the wave train extending further east during the 2015/16 event than the 1982/83 and 1997/98 events (Supplementary Figure S1). We also note a more consistent Rossby wave train occurs during each extreme El Niño event, compared to the wave train during each strong La Niña event (Supplementary Figure S1-S2). The Rossby wave analysis shows that the wave trains during strong La Niña events show greater variability (Supplementary Figure S1-S2). However, the Rossby wave analysis shown here does not allow the full range of mechanisms underpinning our results to be revealed because ENSO-driven Rossby wave trains cannot be easily isolated from local, short term and regional atmospheric circulation and climate changes in the southern midlatitudes (Renwick and Revell 1999; Clem et al. 2018; Yiu and Maycock 2020; McGregor et al. 2022)."

***Comment 3c (cause of differences in SMB):*** *Then, whether the observed difference in the SMB anomaly arises from the difference in ENSO influences or the local variability may become clearer. There might be other ways to distinguish.*

**Response:** As per the previous comments - the impacts of an extreme ENSO event from Rossby wave train propagation from the tropics to the poles unfortunately cannot be isolated from local and regional atmospheric circulation and climatic changes in the southern midlatitudes. The modelling analysis required to separate these influences is outside the scope of this paper.

We are careful to not attribute the presented results to solely ENSO influence and clearly highlight and propose updates to the manuscript to highlight the role of other climate variability and local scale variability (see previous comments above and sections 4.2 in L324-353, 4.4 in L374 - L397).

***Comment 4 (Enderby Land):*** *In Fig. 3, the SLP anomaly appears different for each extreme ENSO. It would be helpful to show moisture fluxes and explain why the SMB anomaly in Enderby Land is the same for all three cases despite different SLP responses. It was only explained by noting that the location is in the center of the action.*

**Response:** We agree that the Enderby Land SLP anomaly is not well explained and further description is needed. This is also needed for the centre of action of the SLP anomaly over Enderby Land.

We propose to include further analysis of the SLP anomaly over Enderby Land during each ENSO event as follows:

L173-175: "During the 2015/16 event, West Antarctic SLP anomalies are similar to the composite, but a negative SLP anomaly extends from Wilkes Subglacial Basin across the Pacific sector (Figure 3m). We note that during both the 1982/83 and 1997/98 events, positive SLP anomalies occur over Enderby Land, which are associated with increased onshore moisture flux from the ocean to the continent (Figure 3e, i). A similar positive SLP anomaly occurs in the composite (Figure 3a). During 1997/98 this SLP over Enderby Land is of much greater magnitude than during other El Niño events and is part of the SLP anomaly that extends across the entire continent (Figure 3i). However, during the 2015/16 event, there is no corresponding positive SLP anomaly over Enderby Land (Figure 3m)."

We also propose to add text that highlights and explains the location of the centre of action in Enderby Land and why this similar response occurs, despite differing SLP responses. We do not provide moisture fluxes in our analysis because Marshall and Thompson (2016) and Marshall et al. (2017) have already examined the Southern Hemisphere polar atmospheric circulation structure, demonstrating how large-scale circulation patterns consistently bring precipitation into the Enderby Land region of East Antarctica and we plan to cite this work. Our suggested new text is below:

L328-332: "Enderby Land is associated with SMB increases during each extreme El Niño event, despite differences in SLP and temperature anomalies between extreme El Niño events. This is because Enderby Land is located within the 'centre of action' of an Indian sector circulation system (Schlosser et al. 2008; Marshall et al. 2016; Marshall et al. 2017) that brings moist, warm air onshore (Schlosser et al. 2008; Marshall et al. 2016; Marshall et al. 2017). Differences in SLP and temperature are associated with this atmospheric circulation system migrating east, west or poleward during each El Niño event (Figure 3).

During the 1982/83 event, a positive SLP anomaly extends from the Lambert-Amery System to Dronning Maud Land (Figure 3e-h). This SLP anomaly shows a weakening of the low-pressure system in the Pacific sector, resulting in moisture-laden, warm air extending across the whole Enderby Land catchment, causing precipitation and SMB increases (Figure 3e-h). During the 1997/98 event, a stronger positive SLP anomaly extends over eastern Dronning Maud Land and across Enderby Land, and the SLP system further? weakens along the Antarctic coastline relative to the 1982/83 event, resulting in an influx of moist air over the west of Enderby Land that drives precipitation and SMB increases in this region (Figure 3i-l). Finally, during the 2015/16 event, a negative SLP anomaly extends across the Pacific sector of East Antarctica, inland and into the Lambert-Amery System, reaching the

interior and eastern Enderby Land. This results from a strengthening and coastward shift of the low-pressure circulation system in the Indian sector (Marshall et al. 2017), causing extensive precipitation across the Lambert-Amery System and the entire catchment of Enderby Land (Figure 3m-p). Therefore, each extreme El Niño event is associated with similar SMB responses, despite differences in the SLP responses (Figure 3)."

***Comment 5 (benefit of using RACMO over ERA5):*** *It is unclear what the benefit of using the regional model is. If the authors draw Figs 3 and 4 from the ERA5 dataset, are they very different? The authors should state the advantages of using the regional model. In particular, the resolutions of both models are not so much different (0.25 degrees vs. 27 km), and the SMB seems to be controlled primarily by moisture transport specified by the boundary conditions.*

**Response:** We thank the reviewer for this suggestion and propose to amend the manuscript and supplement accordingly.

L64-69, we propose to clarify our summary of the RACMO2.3p3 model and add further explanatory text: "We assess Antarctic climate variability using 27 km resolution output from the polar version of the Regional Atmospheric Climate Model version 2.3p3 (RACMO2.3p3) simulated over the period 1979--2018 (van Dalum et al. 2021; 2022). RACMO is more appropriate than ERA5 for addressing SMB impacts due to its finer spatial resolution, consideration of orographic effects, and an updated surface mass balance scheme that includes a firn module (van Dalum et al. 2021; 2022). RACMO has been extensively used to analyse Antarctica's surface climate and previous studies have shown it is accurate and reliable for understanding AIS atmosphere conditions (van Wessem et al. 2014; Leneart et al. 2018; van Wessem et al. 2018; Saunderson et al. 2024; Macha et al. 2024). RACMO2.3p3 is forced at the boundaries by 3-hourly output from the European Centre for Medium-Range Weather Forecasts atmospheric reanalysis (ERA5) (Herbsach et al., 2020; van Dalum et al. 2021) and initialized using a snowpack from a previous model run, meaning that the SMB output for the full time period from 1979–1984 can be utilised (van Dalum et al. 2021). Macha et al. (2024) also affirm this, showing that although 1979–1984 is considered a spin up period in RACMO2.3p3, there is limited impact of excluding this period on the statistical robustness of analyses. We therefore include this period in our analyses. Further detail on RACMO2.3p3 can be found in supplementary text S1. "

L73: "ERA5 atmospheric variables provide insight into the wider circulation anomalies outside the RACMO Antarctic domain, as well as the atmospheric circulation boundaries driving RACMO2.3p3."

In the supplement we propose to include a section providing more detail about RACMO2.3p3, and why it is pertinent to use in this analysis, L59-84: "In this study we utilise RACMO2.3p3, a hydrostatic model developed specifically for use over the polar regions (Carter et al. 2022; van Dalum et al. 2022). RACMO2.3p3 is the updated version of RACMO2.2p2, and includes an updated spectral snow and ice albedo scheme and an updated multi-layer firn module (van Dalum et al. 2022). Surface climate variables in RACMO2.3p3, including SMB, energy balance, surface melt, temperature, albedo and snow grain, have been shown to compare well with both RACMO2.2p2 variables (van Dalum et al. 2022). RACMO also compares well with *in situ* and remotely sensed data, and other ice

sheet model results (van Dalum et al. 2022; Noël et al. 2023; Kappelsberger et al. 2024). Unlike other regional climate models adapted to the Antarctic domain, RACMO2.3p3 represents the insulating properties in the snow column, by including a multi-layer firn module, whilst other models such as the MetUM comparatively utilise a zero-layer snow/soil composite module (Carter et al. 2022; van Dalum et al. 2022).

RACMO2.3p3 couples the High Resolution Limited Area Model version 5.0.3 (HIRLAM) atmospheric dynamics with the European Centre for medium-Range Weather Forecasts (ECMWF) Integrated Forecast System atmospheric and surface physics, using cycle 33rl (ECMWF 2009). In RACMO2.3p3 dry snow metamorphism is calculated using Snow, Ice, and Aerosol Radiation Model, an important component of calculating SMB (Flanner and Zender 2006; Gelman Constantin et al. 2020). RACMO2.3p3 is coupled to the Two-streAm Radiative TransfEr in Snow model (TARTES) through the Spectral-to-Narrow Band Albedo (SNOWBAL) module version 1.2, which allows sub-surface heating and sub-surface melting in the model, both important parts of the ice sheet mass balance and dynamics (Libois et al. 2013). Precipitation is also an important part of SMB calculations in RACMO2.3p3, with fine-scale snow processes and post-depositional accumulation and surface melt processes included in RACMO enabling accurate estimates of SMB in Antarctica (Carter et al. 2022; Nicola et al. 2023; Noël et al. 2023). For these reasons we utilise RACMO2.3p3 near-surface temperature, precipitation and surface mass balance output in our study, as these are adapted to the Antarctic Ice Sheet. ERA5 provides coarser resolution of atmospheric variables which drive RACMO2.3p3 atmospheric circulation boundaries and provide insight into the wider atmospheric circulation anomalies outside the RACMO Antarctic domain (Herbach et al., 2020). ERA5 also does not include a firn module, making its output not as appropriate to address the aims of this study."

***Comment 6 (meaning of statistical significance in Figures 3 and 4):*** *Figures 3 (and 4): Please stress the meaning of statistical significance here. How should one interpret the statistical significance of a single event when each ENSO-induced SMB is so different (including signs in some places) from the others? Or note that the anomaly is significantly different from the baseline but it does not necessarily represent the ENSO response (alone).*

**Response:** Excellent suggestion. We propose a clarification of this in the text associated with Figures 3 and 4, as follows:

L127-129: "Composite anomalies during extreme ENSO events and during each individual El Niño and La Niña event are tested for statistical significance against the climatological baseline conditions, which incorporate all climate variability; therefore, the changes presented are not the result of ENSO alone. "

In L5-9 of the Figure 3 figure caption: "Changes in variables during the SON of each individual extreme El Niño event year and the composite of these events are tested for statistical significance against climatological baseline conditions (which incorporate all climate variability) using a two-tailed Students' *t* test. Non-statistically significant results are shown with stippling at the 5% confidence level. Note that significant changes show a statistically significant difference from the baseline, and are not necessarily representative of a solely ENSO response. "

And in L5-8 of the Figure 4 figure caption: "Changes in variables during the SON of each individual extreme La Niña event year and the composite of these events are tested for statistical significance against climatological baseline conditions using a two-tailed Students' *t* test. Non-statistically significant results are shown with stippling at the 5% confidence level. Note that significant changes show a statistically significant difference from the baseline, and are not necessarily representative of a solely ENSO response."

**Response to Reviewer 2:**

*General comments:*

**Comment 1 (Summary):** *Macha et al., 2024 investigates the effect of extreme ENSO events on the Antarctic surface mass balance using a RCM (RACMO). They found that Antarctic surface mass balance generally does not vary significantly during most extreme ENSO events, though regional differences exist especially over Enderby land (increase in snowfall and SMB); or during the extreme 2015/15 El Niño events.*

*Overall, I find the text well supported by the results where the authors also use robust statistical tests. The figures are clear, and personally I find figure 5 very well done.*

**Response:** We thank the reviewer for their positive feedback and are glad that they find the manuscript to be well supported, with clear figures.

**Comment 2 (summarizing main results):** *In this type of study, it's difficult to balance between an overly detailed analysis by region, which risks losing readers, or something too simple. I sometimes had a little trouble remembering the most important points, even if the balance seems good. I would encourage the authors to summarize the main results at the end of each section to help the reader follow the flow of the manuscript; (and if the results were more coherent for each event, to make a summary graph or table).*

**Response:** Great suggestion. We propose to add summary sentences to the end of each section to help the reader follow the flow of the manuscript, as follows:

L199: "Changes in SLP, temperature, precipitation and SMB across Antarctica are not consistent between the 1982/83, 1997/98 and 2015/16 events, with regional differences in the magnitude and sign of anomalies (Figure 3e-p). The impacts of extreme El Niño events in Antarctica therefore cannot be generalised; the extreme El Niño composite results miss key regional differences in impacts during events (Figure 3a-d). Extreme El Niño event impacts therefore need to be compared on a case-by-case basis."

L221: "There are few similarities between surface climate anomalies, including SLP, surface temperature, precipitation and SMB, during strong La Niña events (Figure 4). The strong La Niña composite results miss key regional differences in the sign and magnitude of climate impacts during individual events (Figure 4). This highlights the importance of considering each event and its impacts individually."

L254: "At the catchment scale, we find that SMB responses vary greatly between individual extreme ENSO events and that there are no consistent SMB responses between extreme El

Niño events, except in Enderby Land, East Antarctica, where SMB anomalies are consistently positive during extreme El Niño events (Figure 5)."

L289: "In summary, most SMB responses during extreme El Niño events are not significantly different from background and average conditions (Figure 6-7). However, SMB responses during the 2015/16 El Niño event are statistically different from both background conditions and moderate ENSO events SMB changes in numerous catchments in both East and West Antarctica (Figure 6-7). Conversely, during strong La Niña events, we do not see evidence of a consistent and significant SMB response (Figure 8). "

***Comment 3 (broader range of El Niño events):*** *Although the authors point this out, I think that the too-small number of events qualifying as extreme reduces the robustness of the analysis. Perhaps it would be possible to use a wider range based on SST distribution percentiles, by varying the threshold defining an extreme event. It might also be the opportunity of using something more statistically-robust than arbitrary of +2 and then -1.5 "because -2 is too limiting for El-Nina events".*

**Response:** We thank the reviewer for this suggestion and have undertaken further analysis of strong El Niño events. We define strong El Niño events following Trenberth et al. (1997) as events with a 1.5°C - 2°C SST anomaly. Strong El Niño events are therefore the positive counterpart of strong La Niña events. However, only two strong El Niño events occurred from 1979–2018: 1987/88 and 1991/92 (pink stars in Supplementary Figure S15). Therefore, we also include moderate El Niño events in our investigation, which we define as events with a 1.0-1.5°C SST anomaly (Trenberth et al., 1997). Four moderate El Niño events occurred during the study period: 1986/87, 1994/95, 2002/03 and 2009/10.

We compare the catchment SMB anomaly for each of these non-extreme El Niño events alongside the results for each extreme El Niño event, CP event and strong La Niña event analysed in our paper (Supplementary Figure S15) to identify key differences. This recreates Figure 5 from the manuscript, with the addition of moderate (purple stars) and strong (pink stars) El Niño events, as can be seen below.

The additional analysis shows that the strong and moderate El Niño events are also associated with a range of SMB responses in each Antarctic catchment. The anomalies vary, with opposite SMB responses occurring in the same catchment during different strong or moderate events. Therefore, the addition of a broader range of El Niño events in our analysis does not change the results and findings from our analysis of solely extreme El Niño events, instead bolstering our results that regional SMB impacts differ between each individual ENSO event and that these impacts cannot be generalised across ENSO events.

Interestingly, Enderby Land, the sole catchment with a consistent positive SMB response to all three extreme El Niño events from our original analysis, also shows a positive SMB anomaly of 15,000kg/m$^2$ during the 1991/1992 strong El Niño event. The Enderby Land 1991/92 SMB response is identified as an outlier in the regions' SMB density curves for SON from 1979-2018.

We thus propose to include text detailing this analysis as follows:

L101: "Moderate and strong El Niño events are also included in our analysis to allow comparison between the impacts associated with extreme and non-extreme El Niño events. Moderate events are defined as events with an SST anomaly between 1°C and 1.5°C for six months or more, whilst strong events are events with an SST anomaly between 1.5°C and 2°C for six months or more (Trenberth et al. 1997)."

L134: "We also assess the cumulative SON SMB anomalies for each moderate or strong El Niño event for each Antarctic catchment, allowing the SMB responses during non-extreme El Niño events to be compared to those during extreme El Niño events (Supplementary Figure S15)."

L250: "We also consider whether moderate and strong El Niño events are associated with more consistent SMB responses. However, as for our analysis of extreme El Niño events, moderate and strong El Niño events do not cause significant and consistent regional SMB changes (Supplementary Figure S15). Interestingly, in Enderby Land (the sole catchment with a consistent positive SMB response to extreme El Niño events) we identify that the 1991/1992 strong El Niño event is associated with a positive SMB anomaly of 15,000kg/m$^2$, which is identified as an outlier in the regions' density curve from Figure 5 (Supplementary Figure S15). "

L306-307: "The addition of strong and moderate El Niño events to our analysis, although increasing the number of events analysed, does not result in a more consistent ENSO signal in SMB. All Antarctic catchments exhibit inconsistent and differing SMB responses between moderate, strong and extreme El Niño events."

L315: "Analysis of moderate and strong El Niño events also identifies that regional SMB changes in Antarctica are indistinguishable during moderate, strong and extreme El Niño events, except in Enderby Land."

[Figure]

**Supplementary Figure S15. Relationship between ENSO events and regional Antarctic surface mass balance anomalies during SON.** Density curves of regional cumulative SON SMB anomalies for each Antarctic Ice Sheet regional catchment, scaled by the regional catchment size. Box plots show the interquartile range (IQR), with medians (black line) and whiskers (5th and 95th percentiles). East Antarctic (light green), West Antarctic (light blue) and Antarctic Peninsula (pink) catchments, outliers (crosses; see supplement), extreme El Niño events (red), strong La Niña events (blue), Central Pacific El Niño events (yellow), strong El Niño events (pink) and moderate El Niño events (purple) are highlighted.

*Comment 4 (SON focus): I also had to reread several times to understand why the manuscript concentrates essentially on the SON period, while all the other periods are somewhat left to the reader's analysis. Even in the supplement, I'd suggest that the authors add a few explanatory lines on what they think is important for the other periods.*

**Response:** We thank the reviewer for bringing this to our attention. We propose to create a separate subsection 2.1.5 named "SON seasonal focus" in the Methods section as follows:

L108-120: "El Niño SST anomalies generally initiate between March and June, and develop in JJA and SON, reaching their peak in DJF, whilst peak La Niña SST anomalies generally occur in SON (Webster, 1982; Ambrizzi et al., 1995; Trenberth, 1997). In this study, we focus on ENSO impacts in SON (of the year when the ENSO event develops), when the ENSO-Antarctic teleconnection is strongest (Webster, 1982; Ambrizzi et al., 1995; Trenberth, 1997) due to a strong subtropical jet enabling Rossby waves to propagate from the tropics to the poles (Renwick and Revell, 1999; Turner, 2004; Yiu and Maycock, 2020). By contrast, the ENSO-Antarctic teleconnection is weakest during DJF, so whilst extreme El Niño events typically peak during DJF, we do not include periods beyond SON in our main analysis (Webster, 1982; Ambrizzi et al., 1995; Trenberth, 1997). For these reasons, all analysis included in the main text is undertaken on SON, and the analysis of other seasons and annual results are included in the supplementary material (Supplementary Figures S1-S3, S9, S10)."

We propose to modify the comparison of seasonal results, with additional text as follows:

L250-254:  "Whilst this manuscript focuses primarily on SON, we also include DJF, MAM and JJA results in Supplementary Figures S1-S3. An analysis of all seasons shows that SMB responses during extreme ENSO events are generally inconsistent, with individual events often exhibiting responses of opposing sign or of differing magnitudes during all seasons (Figure 5; Supplementary Figure S3-S5).

During DJF we identify large positive cumulative SMB anomalies in Dronning Maud Land (2010/11), Enderby Land (1988/89), the Lambert-Amery System (1999/2000) and Princess Elizabeth Land (2007/08; Supplementary Figure S1).  However, there is no consistent response during La Niña events (Supplementary Figure S1).

During both MAM and JJA, we find that there are no consistent responses between extreme El Niño events (comparing red triangles), strong La Niña events (comparing blue triangles), or CP El Niño events (comparing yellow circles; Supplementary Figure S4-S5). During MAM this is unsurprising because ENSO activity during these months tends to be insignificant \citep{trenberthDefinitionNino1997}. During JJA, positive SMB anomalies during different CP El Niño events are identified as outliers in Enderby Land and Lambert-Amery System (Supplementary Figure S5). However, other CP El Niño events are associated with a range of SMB responses in these regions, reinforcing that this is not a consistent response (Supplementary Figure S5)."

Supplementary Text S5, L207-L224: "Supplementary Figures S1-S3 show DJF, MAM and JJA results. These figures show the relationship between extreme ENSO events and regional Antarctic surface mass balance anomalies on a seasonal scale, as they show the SMB anomaly distributions for each Antarctic catchment with each extreme ENSO event highlighted. SMB responses during extreme ENSO events and all seasons are not consistent in East Antarctica, West Antarctica or the Antarctic Peninsula (Supplementary Figure S1-S3).

During DJF, SMB responses are generally inconsistent between events (Supplementary Figure S3). However, during extreme El Niño events SMB anomalies are consistently negative in Enderby Land and the Lambert-Amery System (Supplementary Figure S3). Large positive cumulative SMB anomalies are also identified during austral summer for

different strong La Niña events in Dronning Maud Land (2010/11), Enderby Land (1988/89), the Lambert-Amery System (1999/2000) and Princess Elizabeth Land (2007/08; Supplementary Figure S3). During MAM, no consistent SMB responses are identified during extreme El Niño events, strong La Niña events or CP El Niño events (Supplementary Figure S4). During JJA, SMB responses across Antarctica are generally inconsistent during extreme ENSO events, other than in Enderby Land and Lambert-Amery System during CP El Niño events where positive SMB anomalies are identified as outliers (Supplementary Figure S5)."

**Comment 5a (positive feedback):** *Finally, I have little to reproach this study. The method seems robust to me, the paper remains quite understandable and it's a good piece of work.*

**Response:** Thank you for your constructive feedback and positive review of our manuscript.

**Comment 5b (limitation of study):** *The only rather annoying limitation comes more from the lack of events, and I would really encourage the authors to try to increase the samples in one way or another;*

**Response:** Thanks for this suggestion. We hope that the inclusion of moderate and strong El Niño events addresses this concern. Analysis of paleoclimate data could be another way to increase the number of events in analyses but this is beyond the scope of the current study.

**Comment 5c (improving clarity of paper):** *while taking advantage of the review to further improve the clarity of the paper.*

**Response:** In addition to the above comments, we have worked to improve the clarity of the manuscript, particularly in the results and discussion sections. We propose to reword sentences throughout the manuscript to incorporate details more clearly and remove lengthy sentences and repetition. We propose changes including:

L165: ", that is also identified in the composite (Figure 3a)."

L223: "We next consider whether SMB responses at the catchment scale are distinct from background and average conditions i.e. outside the 5th or 95th percentiles of all SON SMB changes during the 1979–2018 period."

L307: "Moderate and strong La Niña events are also associated 390 with a range of inconsistent SMB responses during these events. However when extreme El Niño event SMB changes are compared to SMB changes during moderate El Niño events, SMB responses are significantly different from one another in Enderby Land, Wilkes Subglacial Basin, Victoria Land, Ross East, Ross West, Getz, Amundsen Sea, Abbott, George VI and Ronne catchments."

**Comment 6 (missing spaces):** *PS: there are two missing spaces L47 and 49.*

**Response:** Amended.

**References**

Cai, W. et al. 2014. Increasing frequency of extreme El Niño events due to greenhouse warming. *Nature Climate Change* 4(2), pp. 111–116. doi: 10.1038/nclimate2100.

Capotondi, A. et al. 2015. Understanding ENSO Diversity. *Bulletin of the American Meteorological Society* 96(6), pp. 921–938. doi: 10.1175/BAMS-D-13-00117.1.

Carter, J., Leeson, A., Orr, A., Kittel, C. and Van Wessem, J.M. 2022. Variability in Antarctic surface climatology across regional climate models and reanalysis datasets. *The Cryosphere* 16(9), pp. 3815–3841. doi: 10.5194/tc-16-3815-2022.

Clem, K.R., Renwick, J.A. and McGregor, J. 2018. Autumn Cooling of Western East Antarctica Linked to the Tropical Pacific: ENSO and East Antarctic climate. *Journal of Geophysical Research: Atmospheres* 123(1), pp. 89–107. doi: 10.1002/2017JD027435.

van Dalum, C.T., van de Berg, W.J. and van den Broeke, M.R. 2022. Sensitivity of Antarctic surface climate to a new spectral snow albedo and radiative transfer scheme in RACMO2.3p3. *The Cryosphere* 16(3), pp. 1071–1089. doi: 10.5194/tc-16-1071-2022.

Kappelsberger, M.T. et al. 2024. How well can satellite altimetry and firn models resolve Antarctic firn thickness variations? *The Cryosphere* 18(9), pp. 4355–4378. doi: 10.5194/tc-18-4355-2024.

L'Heureux, M.L. et al. 2017. Observing and Predicting the 2015/16 El Niño. *Bulletin of the American Meteorological Society* 98(7), pp. 1363–1382. doi: 10.1175/BAMS-D-16-0009.1.

Libois, Q., Picard, G., France, J.L., Arnaud, L., Dumont, M., Carmagnola, C.M. and King, M.D. 2013. Influence of grain shape on light penetration in snow. *The Cryosphere* 7(6), pp. 1803–1818. doi: 10.5194/tc-7-1803-2013.

Marshall, G.J. and Thompson, D.W.J. 2016. The signatures of large-scale patterns of atmospheric variability in Antarctic surface temperatures. *Journal of Geophysical Research: Atmospheres* 121(7), pp. 3276–3289. doi: 10.1002/2015JD024665.

Marshall, G.J., Thompson, D.W.J. and Broeke, M.R. 2017. The Signature of Southern Hemisphere Atmospheric Circulation Patterns in Antarctic Precipitation. *Geophysical Research Letters* 44(22), p. 11,580-11,589. doi: 10.1002/2017GL075998.

McGregor, S., Cassou, C., Kosaka, Y. and Phillips, A.S. 2022. Projected ENSO Teleconnection Changes in CMIP6. *Geophysical Research Letters* 49(11). Available at: https://onlinelibrary.wiley.com/doi/10.1029/2021GL097511 [Accessed: 26 June 2023].

Nicola, L., Notz, D. and Winkelmann, R. 2023. Revisiting temperature sensitivity: how does Antarctic precipitation change with temperature? *The Cryosphere* 17(7), pp. 2563–2583. doi: 10.5194/tc-17-2563-2023.

Noël, B., Van Wessem, J.M., Wouters, B., Trusel, L., Lhermitte, S. and Van Den Broeke, M.R. 2023. Higher Antarctic ice sheet accumulation and surface melt rates revealed at 2 km resolution. *Nature Communications* 14(1), p. 7949. doi: 10.1038/s41467-023-43584-6.

Renwick, J.A. and Revell, M.J. 1999. Blocking over the South Pacific and Rossby Wave Propagation. *Monthly Weather Review* 127(10), pp. 2233–2247. doi: 10.1175/1520-0493(1999)127<2233:BOTSPA>2.0.CO;2.

Santoso, A., Mcphaden, M.J. and Cai, W. 2017. The Defining Characteristics of ENSO

Extremes and the Strong 2015/2016 El Niño. *Reviews of Geophysics* 55(4), pp. 1079–1129. doi: 10.1002/2017RG000560.

Timmermann, A. et al. 2018. El Niño–Southern Oscillation complexity. *Nature* 559(7715), pp. 535–545. doi: 10.1038/s41586-018-0252-6.

Welhouse, L.J., Lazzara, M.A., Keller, L.M., Tripoli, G.J. and Hitchman, M.H. 2016. Composite Analysis of the Effects of ENSO Events on Antarctica. *Journal of Climate* 29(5), pp. 1797–1808. doi: 10.1175/JCLI-D-15-0108.1.

Xue, Y. and Kumar, A. 2017. Evolution of the 2015/16 El Niño and historical perspective since 1979. *Science China Earth Sciences* 60(9), pp. 1572–1588. doi: 10.1007/s11430-016-0106-9.

Yiu, Y.Y.S. and Maycock, A.C. 2020. The linearity of the El Niño teleconnection to the Amundsen Sea region. *Quarterly Journal of the Royal Meteorological Society* 146(728), pp. 1169–1183. doi: 10.1002/qj.3731.

---

## Author Response (AR2)

February 2025

**Authors response to reviews on '*How do extreme ENSO events affect Antarctic surface mass balance?*' submitted to *The Cryosphere.**

Masashi Niwano
Editor, The Cryosphere

Thank you for considering our revised manuscript.

We are pleased that Anonymous Referee #1 is generally satisfied with our responses, with only two minor revisions remaining. We are also pleased the Editor has found our responses to Christoph Kittel (Referee #2) to be convincing. We thank Anonymous Referee #1 for their continued feedback, comments and suggestions which will further improve the manuscript.

Here, we provide responses to each of the reviewers' comments. Reviewer comments are in *italics.* Our responses are included in regular text. When noting our changes, we refer to both line numbers in the original manuscript and include wording changes in blue text and quotation marks (" ").

Jessica Macha and co-authors

**Response to Reviewer 1:**

**General comments:**
**Comment:** I appreciate the authors' great effort in improving the manuscript in response to my earlier comments. In particular, the careful writing and reserved attribution of the SMB anomaly to ENSO, taking other possibilities into account. The analysis of Rossby wave trains is also useful to identify where the differences originate. I still have two comments, of which the first one is important.

**Response:** We thank the reviewer for their constructive feedback and are glad that they appreciate the improved manuscript, including the clarification of reserved attribution, and the Rossby wave train analysis.

**Specific Comments:**
**Comment 1a:** Throughout the text, the authors use the term "impacts of El Niño". As the authors now agree, what is shown is not necessarily the impact of El Niño alone. It is the SMB anomaly during the El Niño event. For example, it says in the abstract (L.6) that "Regional impacts differ between individual events and cannot be generalized across all extreme events." More accurately, this should be "regional anomalies", rather than "regional impacts", as the causality is not demonstrated. I notice similar usage of "impacts" throughout the text. I suggest re-examining each use of the term "impacts".

**Response:** Yes, we agree that the use of "impacts of El Niño" is ambiguous as we have clarified that what is shown is not necessarily the impact of El Niño alone. Therefore as suggested we have re-examined the use of the term 'impacts' throughout the manuscript and reworded the following sentences (below):

Abstract L6: "Based on only three (five) events in the observational period, regional anomalies differ during the extreme El Niño (La Niña) events considered and cannot be generalised."

L56: "Overall, we aim to answer the following questions: do Antarctic SMB anomalies during extreme ENSO events follow a similar pattern? Where do these impacts occur? And more generally, do these extreme ENSO events also result in extreme Antarctic SMB changes?"

L107: "Non-extreme CP El Niño events (see section 2.1.3) are also included in our analysis, to provide a comparison to extreme El Niño events, building off Macha et al. (2024), which found that CP-type El Niño events result in widespread SMB increases in West Antarctica."

L109: "Moderate and strong El Niño events are also included in our analysis to allow comparison with El Niño events of lower magnitude."

L122: "In this study, we focus on ENSO in SON (of the year when the ENSO event develops), when the ENSO-Antarctic teleconnection is strongest"

L233: "Strong La Niña events induce a range of surface climate changes, with no clear pattern in common between strong La Niña events SLP, temperature, precipitation and SMB anomalies (Figure 4)."

L360: "Here, we have considered Antarctic SMB anomalies during extreme ENSO events in the historical record to determine if extreme ENSO events are associated with extreme SMB changes in Antarctica."

L365: "However, beyond the 2015/16 event and Enderby Land, our results show that extreme changes in Antarctic climate and SMB do not occur during extreme ENSO events."

Section 4.1 title: "Extreme ENSO events and moderate ENSO events"

L393: " However these results are not consistent across all three extreme El Niño events studied (other than in Enderby Land)."

L395: "Conversely, numerous other catchments exhibit similar SMB responses when extreme and moderate El Niño event SMB changes are compared. That is, the SMB changes during extreme ENSO events do not differ from those during moderate events

in Dronning Maud Land, Lambert-Amery system, Princess Elizabeth Land, Aurora Subglacial Basin, West Graham Land, Larsen, Filchner and Coats catchments."

L449: "Our findings further show that the 2015/16 event stands out relative to previous events, and is associated with more widespread and significant Antarctic SMB changes than during other extreme ENSO events."

L458: "This magnitude difference was partially attributed to the 2015/16 event being initiated from a warmer tropical Pacific background state than the 1982/83 and 1997/98 events (Santoso et al., 2017), resulting in the higher magnitude and more widespread anomalies in Antarctica."

L464: "Attributing whether there is an anthropogenic signal in these extreme ENSO events and SMB anomalies is beyond the scope of this study and requires centennial scale datasets to fully characterise ENSO variability (Stevenson et al., 2010)."

L488: "Our analysis of Antarctic surface mass balance during extreme ENSO events is limited by the length of the datasets available."

*Comment 1b:* While I partially understand the authors' intention, I think the following statements are too strong and can be misleading: "The impacts of extreme El Niño events in Antarctica therefore cannot be generalised; the extreme El Niño composite results miss key regional differences in impacts during events (Figure 3a-d). Extreme El Niño event impacts therefore need to be compared on a case-by-case basis."

By these statements, the authors implicitly attribute the observed anomalies to the extreme El Niño events. As the authors now carefully phrased in places throughout the text, the differences are not attributable to El Niño alone. Therefore, there is no evidence that "El Niño event impacts need to be compared on a case-by-case basis".

The composite analysis (or arithmetic average) is a powerful statistical tool to generalize the phenomena but has little meaning with a few samples. I think "The impacts of extreme El Niño events in Antarctica" can "be generalized" if there are enough samples. This is not what we have learned from this study but is supported by the textbook. Therefore, there is no convincing evidence that "The impacts of extreme El Niño events in Antarctica therefore cannot be generalized".

I also disagree with the point written in the response that "a key finding of our analysis is that the composite analysis cannot be used to generalize the behaviour of extreme El Nino or La Nina events." This is not necessarily a finding from this study but may be a statistical limitation.

It is safe to argue that the individual extreme events are statistically different from the baseline (because that is what the extremes are), but it is difficult to make a statistical statement about commonality or dis-commonality among the three samples. In my

opinion, this is a very valuable study consisting of three individual cases, but not more than that.

**Response:** We thank the reviewer for raising this point. We agree that the statements highlighted are too strong in wording, and implicitly attribute the observed anomalies to the extreme El Niño events. We also agree that the three individual extreme events are statistically different from the baseline, and that our findings do not extend further than these individual events. We therefore have reworded these points in the manuscript, as follows:

Abstract, L6: "Based on only three (five) events in the observational period, regional anomalies differ during the extreme El Niño (La Niña) events considered and cannot be generalised."

L229: "Changes in SLP, temperature, precipitation and SMB across Antarctica are not consistent between the 1982/83, 1997/98 and 2015/16 events, with regional differences in the magnitude and sign of anomalies (Figure 3e-p). The composite of extreme El Niño events also appears to miss key regional differences between events; however, this may be related to the small number of events in the analysis period (Figure 3a-d). Given this limitation, in this study we compare extreme El Niño events on a case-by-case basis to ensure that differences between events are adequately accounted for."

L254: "There are few similarities between surface climate anomalies, including SLP, surface temperature, precipitation and SMB during strong La Niña events (Figure 4e-x). The composite of strong La Niña events also appears to miss regional differences in the sign and magnitude of climate anomalies during individual events (Figure 4a-d). As for the El Niño composites, this may be due to statistical limitations associated with the small number of events considered. When conducting extreme ENSO event analysis in this study, each event and its impacts are assessed, to ensure key differences between events or impacts are not overlooked."

L316: "At the catchment scale, we find that SMB responses vary greatly between individual extreme ENSO events and that there are no consistent SMB responses between the three extreme El Niño events considered, except in Enderby Land, East Antarctica, where SMB anomalies are consistently positive (Figure 5)."

L354: "In summary, most SMB responses during the three extreme El Niño events identified are not significantly different from background and average conditions (Figure 6-7)."

**Comment 1c:** It is safe to argue that the individual extreme events are statistically different from the baseline (because that is what the extremes are), but it is difficult to make a statistical statement about commonality or dis-commonality among the three samples. In my opinion, this is a very valuable study consisting of three individual cases, but not more than that.

**Response:** We thank the reviewer for raising this point. We agree that the individual extreme events are statistically different from the baseline.

*Comment 2:* Could the author check whether the following statement is accurate? "RACMO is more appropriate than ERA5 for addressing SMB impacts due to its finer spatial resolution…" The ERA5 dataset is provided at 0.25x0.25 resolution (not the same as the original numerical model resolution), and 0.25 degree is about 27 km at the equator (same as RACMO). If one goes to polar regions, the zonal resolution becomes much finer. This is the reason why I originally asked the author if RACMO has the advantage of simulating moisture transport in the polar region than ERA5.

**Response:** Thank you for highlighting this inaccuracy. The reviewer is correct, we have stated that RACMO is of finer spatial resolution than ERA5 which is inaccurate. However, as outlined in detail in Supplementary Text S1, there are extra processes included in RACMO that improve the representation of SMB, resulting in more accurate Antarctic simulations of SMB than utilising P-E in ERA5. We have updated this text to correct the inaccuracy around spatial resolution and highlight the content of Supplementary Text S1 more clearly.

L67-68: "RACMO is more appropriate than ERA5 for addressing SMB impacts due to its adaptation to the polar regions, including consideration of orographic effects, post-depositional processes and an updated surface mass balance scheme that includes a firn module (van Dalum et al. 2022). We include a more detailed discussion on the polar developments in  RACMO2.3p3, and justification for its use over ERA5, in Supplementary Text S1."